# Discoidin-domain receptor coordinates cell-matrix adhesion and collective polarity in migratory cardiopharyngeal progenitors

Yelena Y. Bernadskaya[1], Saahil Brahmbhatt[1], Stephanie E. Gline[1], Wei Wang[1] & Lionel Christiaen [1]

Integrated analyses of regulated effector genes, cellular processes, and extrinsic signals are required to understand how transcriptional networks coordinate fate specification and cell behavior during embryogenesis. *Ciona* cardiopharyngeal progenitors, the trunk ventral cells (TVCs), polarize as leader and trailer cells that migrate between the ventral epidermis and trunk endoderm. We show that the TVC-specific collagen-binding Discoidin-domain receptor (Ddr) cooperates with Integrin-β1 to promote cell-matrix adhesion. We find that endodermal cells secrete a collagen, Col9-a1, that is deposited in the basal epidermal matrix and promotes Ddr activation at the ventral membrane of migrating TVCs. A functional antagonism between Ddr/Intβ1-mediated cell-matrix adhesion and Vegfr signaling appears to modulate the position of cardiopharyngeal progenitors between the endoderm and epidermis. We show that Ddr promotes leader-trailer-polarized BMP-Smad signaling independently of its role in cell-matrix adhesion. We propose that dual functions of Ddr integrate transcriptional inputs to coordinate subcellular processes underlying collective polarity and migration.

---

[1] Center for Developmental Genetics, Department of Biology, New York University, New York 10003 NY, USA. Correspondence and requests for materials should be addressed to L.C. (email: lc121@nyu.edu)

During embryonic development, complex tissue-scale movements emerge from coordinated behaviors of individual cells. Morphogenesis is largely tissue-specific, indicating that gene regulatory networks (GRNs) that control cell identity also determine cell behavior. How GRNs control and coordinate cell fate and behavior has been illustrated using diverse models such as mesoderm invagination and migration in Drosophila and sea urchins, gastrulation in Xenopus, and neural crest cell migration in amniotes[1–8]. Collective migration is observed in various physiological and pathological conditions such as neural crest cell migration, gastrulation, wound healing, and cancer metastasis[9,10]. During collective movements, cells of the same identity adopt leader and follower states with distinctive morphologies[11]. Collective polarity is established, maintained and coordinated with the direction of movement, and polarity and directionality arise from anisotropic exposure to extrinsic cues, such as free edges of leading cells, gradients of secreted molecules from surrounding tissues, or asymmetric distribution of adjacent extracellular matrix[12,13]. Integrative mechanisms therefore connect regulatory inputs with production of state-specific cell behavior in response to extrinsic cues.

The cardiac lineage in the tunicate Ciona provides the simplest example of collective cell migration[13–16]. In Ciona, multipotent cardiopharyngeal progenitors derive from the B7.5 blastomeres in 110-cell embryos, and later produce heart and pharyngeal muscle precursors[17–20]. Bilateral pairs of cardiopharyngeal progenitor cells (aka trunk ventral cells, TVCs) collectively polarize and migrate between the ventral epidermis and trunk endoderm (Fig. 1a), until they stop and divide asymmetrically to produce fate-restricted progenitors[13,14,16,19]. During migration, the leader TVC extends dynamic protrusions, generating a broad leading edge, while the trailer terminates in a tapered retracting end[13,16]. Prior to migration, the surrounding trunk endoderm preferentially contacts the prospective leader cell, and experimental perturbation of secretion in endoderm cells disrupts collective leader-trailer (LT) polarity[13], suggesting that anisotropic exposure to extrinsic endodermal cues contributes to collective TVC polarity. Moreover, B7.5-lineage-specific transcriptional inputs from Mesp, FGF/MAPK signaling and Foxf control and coordinate cardiopharyngeal fate specification and TVC migration[15,16,18,21], and the transcriptome of migratory TVCs has been extensively profiled[16,22–24]. Therefore, TVC migration provides an attractive model to functionally connect fate-specific transcriptional inputs with cellular effectors and extrinsic cues governing collective polarity and migration.

Among regulated cellular effectors possibly shaping cell responses to extrinsic cues, we identified several receptor tyrosine kinases (RTKs)[16]. Other developmental cell behaviors involve RTK signaling, including collective polarization of migratory cell groups in vertebrates and Drosophila[10,25–30]. We selected a small group of candidate RTKs and analyzed their roles in TVC migration. Among these candidates, the sole Ciona homolog of Discoidin domain receptor (Ddr) is upregulated specifically in newborn TVCs, downstream of FGF/MAPK signaling and Foxf inputs[16]. DDRs are single-pass transmembrane receptors that bind extracellular collagen[31,32], mediate weak collagen adhesion, and regulate cadherins, integrin, and ECM interacting proteins to modulate cell-matrix adhesion[33–35].

Here we describe the functions of Ddr and its interactions with integrin, Vegfr, and BMP-Smad signaling pathways in regulating cell-matrix adhesion and collective polarity during TVC migration. We developed methods to quantify TVC morphology and movements, and defined an experimental and analytical framework to study morphogenetic determinants. We find that the endoderm secretes a type IX collagen, Col9-a1, that is deposited onto the basal epidermal matrix, contributing to activation of Ddr on the ventral surface of the migrating TVCs, thus promoting integrin-based cell-matrix adhesion. We dissected a signaling antagonism between Ddr/Integrin-mediated cell-matrix adhesion and Vegfr signaling. We show that Ddr promotes polarized BMP-Smad signaling, thus contributing to establishing distinct leader and trailer states. The latter function appears independent from integrin-mediated cell-matrix adhesion, suggesting that dual functions of Ddr coordinate collective polarity and cell-matrix adhesion during TVC migration.

## Results

**Cardiopharyngeal progenitors express RTKs.** Transcriptome profiling identified signaling molecules potentially involved in guiding movements of Ciona cardiopharyngeal progenitors, the TVCs. We used whole mount in situ hybridization to characterize the expression of four candidate RTK-coding genes, Discoidin domain receptor (Ddr), Vascular endothelial growth factor receptor (Vegfr), Fibroblast growth factor receptor (Fgfr), and epidermal growth factor receptor (Egfr). No Egfr transcripts were detected prior to or during TVC migration, neither did overexpression of truncated dominant negative forms produce detectable TVC migration phenotypes (Supplementary Figure 1, Supplementary Movie 5). We thus excluded Egfr from further analysis. Ddr, Vegfr, and Fgfr transcripts were detected in migrating TVCs (Fig. 1b, Supplementary Figure 1). Vegfr and Fgfr were expressed in B7.5 lineage founder cells, while newborn TVCs upregulated Ddr before the onset of collective migration (Supplementary Figure 1).

Foxf is proposed to act as a key transcriptional regulator of TVC migration[15], and transcription profiling identified candidate Foxf target genes, including Ddr[16]. Using B7.5-lineage-specific CRISPR/Cas9-mediated mutagenesis with single guide RNAs (sgRNAs;[36]) targeting Foxf, we confirmed that loss of Foxf expression inhibited Ddr expression in the TVCs (Fig. 1b, c). By contrast, Foxf appeared dispensable for Vegfr or Fgfr expression, a finding consistent with previous microarray analyses[16]. Parallel studies indicate that Fgfr is primarily required for MAPK-dependent transcriptional regulation in migrating TVCs and beyond[16,21,37]. These data led us to focus on Ddr and Vegfr as candidate cell migration effectors, which we sought to further characterize.

**RTK functions required for proper TVC migration.** TVC migration is stereotypical in control embryos. Pairs of cousin TVCs collectively polarize to assume leader and trailer positions, potentially through differential contacts with the mesenchyme and trunk endoderm[13], and these relative positions are maintained throughout migration. The cells move anteriorly, between the ventral epidermis and trunk endoderm[13,16], until they reach a final position adjacent to the midline and divide asymmetrically to generate the first and second heart precursors, and atrial siphon muscle precursors[19,20]. To characterize TVC migration in control and experimental conditions, we developed quantitative parameters describing cell movements in live embryos. We used an epidermal transgene, EfnB > hCD4::mCherry[13], and a B7.5-lineage-specific nuclear marker, Mesp > H2B::GFP, to label migrating cardiopharyngeal progenitors and epidermal cells in live embryos and imaged them using time lapse confocal microscopy for the duration of TVC migration (Fig. 1d, Supplementary Movies 1–5). To quantify TVC movements in four dimensions, we used morphological landmarks in the epidermis to define sagittal and frontal planes. We used GFP-positive TVC nuclei to define the leader-trailer (LT) axis, and calculated the angles that axis formed with sagittal and frontal planes at each time point (Fig. 1d, Supplementary Figure 2A, Methods). We

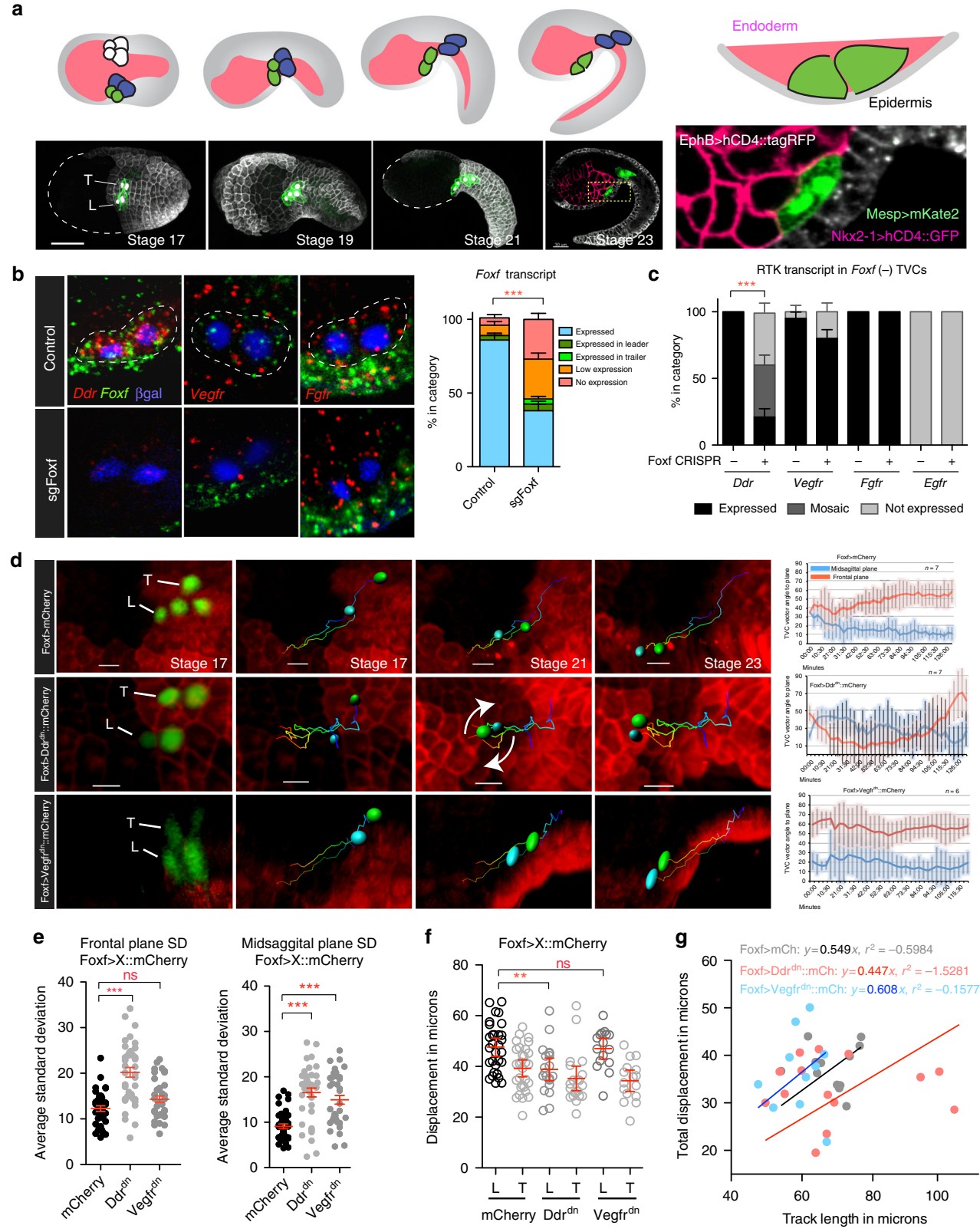

quantify the variability of cell positions by calculating the standard deviation of the angles between leader-trailer axes and the sagittal and frontal planes at each time point (Fig. 1d), assuming that migrating cells that are constrained during their movement will have a narrow SD profile. We used cell division times to align

temporal axes and combine time-lapse recordings from multiple embryos ($n = 7$). We found that the TVC migration path is constrained in control embryos, with LT angles with frontal and sagittal planes only changing within defined ranges as cells migrate (Fig. 1e, f, Supplementary Figure 2).

**Fig. 1** Ddr and Vegfr regulate TVC migration. **a** Schematic of TVC migration between endoderm and epidermis. TVCs (green), ATMs (blue), endoderm (pink), and epidermis (gray). Lower panel shows micrographs of TVC migration during development. Epidermal cells are marked with *EfnB > hCD4::mCherry*, B7.5 lineage marked with *Mesp > hCD4::GFP* and nuclear *Mesp > H2B::mCherry* for stages 17–19, markers used for 3-channel acquisition at stage 23 are given on magnified image. Magnified image shows the positioning of the TVCs between the endoderm and epidermis. Scale bar = 60um. **b** Fluorescent in situ hybridization (FISH) in migrating TVCs under control (Ebf$^{CRISPR}$, $n = 101$) and CRISPR ($n = 118$) targeting the *Foxf* locus. Micrographs are oriented with leader TVC to the left. TVC pair outlined with dotted lines and nuclei marked with *Mesp > LacZ* and stained for β-galactosidase. Quantification of *Foxf* knockout efficiency is shown to the right. Standard error of proportion is shown, statistical analysis using the $X^2$ test. **c** RTK mRNA expression in proportion of embryos that lose *Foxf* expression. sgRNAs targeting the EBF locus used as control. Standard error of proportion and statistical significance is shown using the $X^2$ test. For **b**, **c**, the data are pooled from three biological replicates. **d** TVC migration path tracking. TVC and ATM nuclei are marked with *Mesp > H2B::GFP* and epidermis with *EfnB > hCD4::mCherry*. Tracks of migrating TVCs shown in each panel, cyan sphere = leader, green sphere = trailer. Arrows highlight tumbling of TVCs. Average angle of L/T axis to sagittal and frontal planes are shown with standard deviation. **e** Average standard deviation of TVC position at the sagittal and frontal planes. Statistical analysis using one-way ANOVA. S.E.M. is shown. **f** Average displacement of leader/trailer TVCs measured from anterior ATM. Statistical analysis using two-way ANOVA and Bonferroni post-test. 95% C.I. is shown. The data are pooled from two biological replicates. **g** Relationship of total TVC displacement and track length. The data are derived from live imaging shown in subset **d**. For the regression, line is forced to go through the origin. Slope of the line represents track straightness calculated as displacement divided by total track length. For all graphs, *$p < 0.05$, **$p < 0.005$, ***$p < 0.0005$. L leader, T Trailer

To study the functions of selected RTKs during TVC migration, we generated passive dominant negative versions of Ddr and Vegfr (henceforth referred to as Ddr$^{dn}$ and Vegfr$^{dn}$), by deleting their cytoplasmic kinase domains[38]. We expressed these constructs in the TVCs using a defined *Foxf* enhancer[15], and assayed migration quantitatively. TVCs expressing *Foxf*-driven Ddr$^{dn}$ retained their ability to initiate migration, but displayed more variable relative leader/trailer positions, occasionally resulting in a tumbling motion (15%; $n = 7$; Fig. 1d, e; Supplementary Movie 2). Ddr$^{dn}$ misexpression also caused a significant decrease in final displacement of TVCs, an effect that was not observed in TVCs expressing Vegfr$^{dn}$ (Fig. 1f). The decreased migration distance observed using fixed samples under Ddr$^{dn}$ conditions coincides with a reduction of track straightness observed in live imaging, with the cells traveling the least straight path also having smaller displacement, consistent with altered directionality (Fig. 1g). These observations suggest that proper function of the Foxf target Ddr is required for maintenance of collective polarity and directional migration of TVCs.

TVCs expressing *Foxf*-driven Vegfr$^{dn}$ were also able to initiate migration, and their total displacement was comparable to control cells (Fig. 1f). However, Vegfr$^{dn}$ expression slightly increased track straightness, which potentially reflects an increased canalization during migration (Fig. 1g). We compared the variability of TVC positions by comparing the standard deviation of angles to frontal or sagittal planes assumed by the TVCs at each time point during migration. As expected, Ddr$^{dn}$ increased the standard deviation of the LT angle relative to both the sagittal and the frontal plane, reflecting the highly variable positions of tumbling TVCs. By contrast, Vegfr$^{dn}$ only increased the standard deviation of the LT angle relative to the sagittal plane, which is consistent with cells remaining constrained in at least one dimension (Fig. 1e).

**Ddr promotes integrin-based adhesion to the epidermis**. We focused on understanding how Ddr affects TVC morphology and contacts with surrounding tissues[13]. Using defined transgenes to label TVC membranes as well as the underlying epidermis, we used confocal imaging and computational segmentation of individual cells to quantify morphometric parameters, including cell sphericity and percentage of total cell surface contacting the epidermis (Fig. 2a). By revealing quantitative differences between leader and trailer cells, these measurements allowed us to characterize collective polarity in control and experimental conditions.

Leader TVCs are less spherical than the trailers, consistent with their greater protrusive activity[16] (Fig. 2b). Approximately

45–50% of the leader surface contacts the underlying epidermis compared to 40–45% for the trailer (Fig. 2a, c). Expression of Ddr$^{dn}$ significantly increased the sphericity of the leader, abolishing leader-trailer differences (Fig. 2a, b). Concurrently, Ddr$^{dn}$ significantly reduced the surface of TVC contact with the underlying epidermis in a dose-dependent manner (Fig. 2a, c; Supplementary Figure 3). Ddr$^{dn}$-expressing TVCs decreased epidermal contact of both leader and trailer to ~30% of their surface on average, with 75% of the cells assayed displaying variable contact with the epidermis and ~15% becoming fully detached from the epidermis (Fig. 2c, Supplementary Figure 3). Complementary loss-of-function approaches validated the specificity of the Ddr$^{dn}$-induced phenotype, as lineage-specific RNAi using short hairpin microRNA (shmiR) constructs showed a trend suggesting synergy with low doses of Ddr$^{dn}$ to cause a penetrant de-adhesion phenotype (Supplementary Figure 4). We observed more penetrance and reproducibility of the de-adhesion phenotype using the dominant negative form of Ddr and therefore continue to use it in all subsequent analysis. These data suggest that Ddr function is required for TVCs to maintain contact with the epidermis during migration.

Phenotypes produced by Ddr$^{dn}$ suggested that cells failed to maintain adhesion to the extracellular matrix presumably lining the ventral epidermis. Ddr homologs can mediate weak adhesion to collagen and increase integrin-mediated adhesion to collagen-based extracellular matrices[34,39]. We sought to identify Ddr's integrin partners in the migrating TVCs. Using whole mount fluorescent in situ hybridization assays and TVC-specific transcriptome profiling data[16], we assayed multiple Integrin-β transcripts for TVC expression. Among several candidates, only Integrin-β1 (*Intβ1*) mRNA was detectable in TVCs prior to and during migration (Supplementary Figure 6). We generated a dominant negative form of Intβ1 (Intβ1$^{dn}$) by truncating the C-terminal cytoplasmic domain[40], and assayed cell sphericity and the ability of Intβ1$^{dn}$-expressing TVCs to contact the epidermis, to test whether it would phenocopy Ddr$^{dn}$ misexpression (Fig. 2a–c). Altering Intβ1 function increased cells' sphericity to levels comparable to those observed with Ddr$^{dn}$ and abolished the sphericity difference between leader and trailer (Fig. 2b). Furthermore, Intβ1$^{dn}$ misexpression reduced the TVCs' epidermal contact to ~35% of their surface (Fig. 2c). Compared to control cells, this reduction is similar to that observed with Ddr$^{dn}$. Finally, Intβ1$^{dn}$ also increased the variability of TVC position during migration, similar to Ddr$^{dn}$ (Fig. 2e, f). These observations indicate that Intβ1$^{dn}$ mimics the Ddr$^{dn}$ misexpression phenotype, and reveal a trend where the cells with the highest sphericity have the least surface devoted to contact with the underlying epidermis (Fig. 2h).

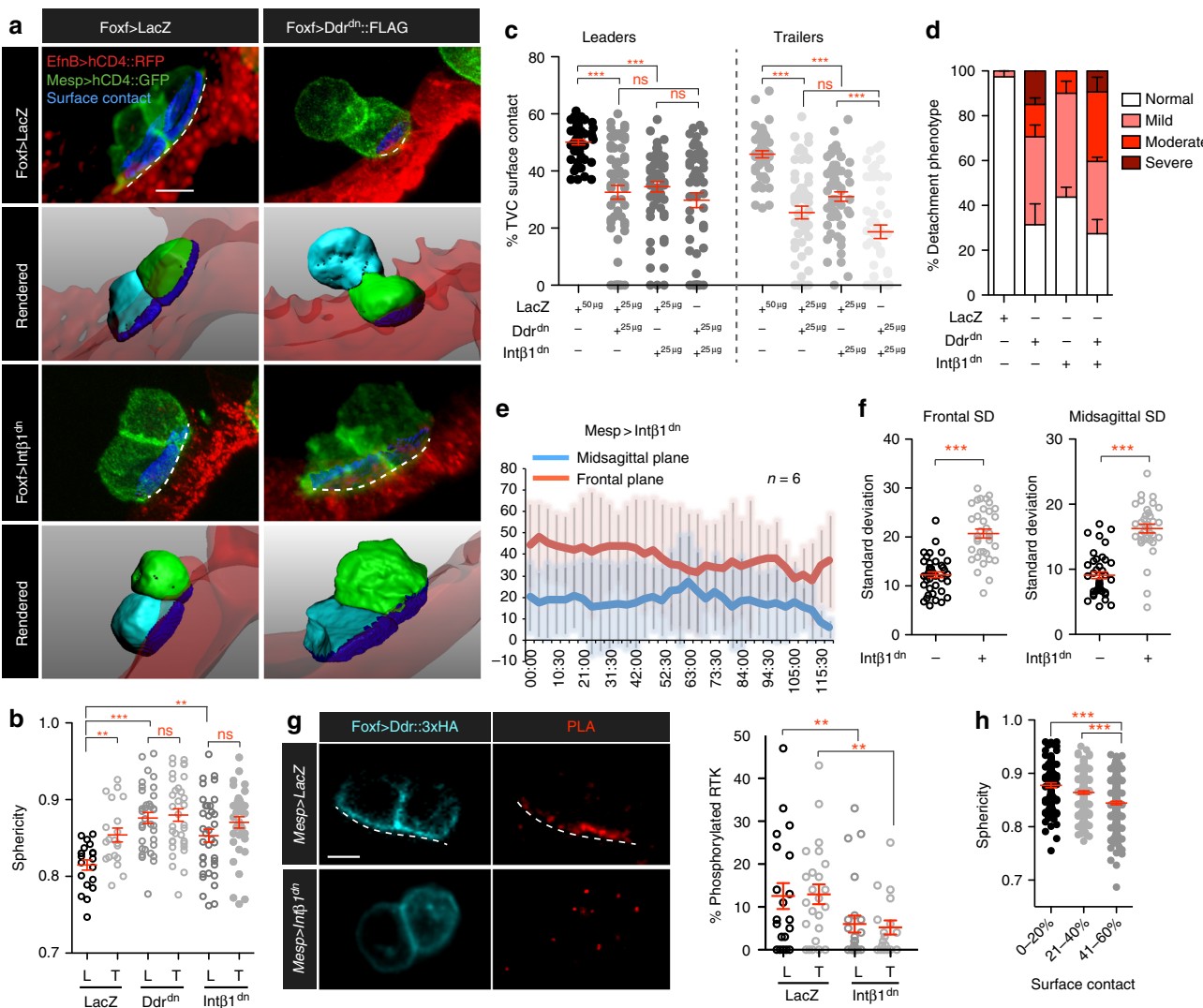

**Fig. 2** Integrin-β1 and Ddr promote TVC adhesion to the ventral epidermis. **a** Micrographs of surface contacts made by migrating TVCs with the underlying epidermis. Surface contacts are shown in blue. The length of the surface contact is indicated with a dashed line. TVCs are marked and segmented based on the B7.5 linage-specific expression of *Mesp > hCD4:GFP* and epidermal surface is visualized with *EfnB > hCD4::RFP*. Lower panels show rendered surfaces, leader is shown in cyan, trailer in green, epidermal surface in transparent red. For segmentation and derivation of surface contacts see Materials and Methods. Scale bar = 10 μm. **b** TVC sphericity under adhesion perturbation. The data are pooled from three biological replicates. Statistical analysis using two-way ANOVA followed by the Bonferroni post test. S.E.M. is shown. **c** TVC surface contact under adhesion perturbation. All transgenes are expressed from the Foxf TVC-specific enhancer. Results shown are pooled from 3 biological replicates. Statistical analysis using two-way ANOVA followed by the Bonferroni post hoc test. S.E.M. is shown. **d** Interaction of Ddr and Intβ1 adhesion pathways. Bar graph shows percent detachment phenotype detected under each condition. Standard error of proportion is shown. **e** Quantitation of migration angles of *Mesp > Intβ1dn* expressing TVCs, averages at each time point and standard deviation shown. **f** Average standard deviation relative to the frontal and sagittal planes of *Mesp > Intβ1dn* expressing TVCs. Statistical analysis using the Wilcoxon rank sum test. S.E.M. is shown. **g** Proximity ligation assay (PLA) of phosphorylated full-length Ddr under control and Intβ1dn conditions. Quantitation of phosphorylated Ddr compared to the total expression of Ddr. Statistical analysis using a two-way ANOVA followed by the Bonferroni post hoc test. S.E.M is shown. Scale bar = 10 μm. **h** Changes in sphericity associated with reduction in TVC/epidermis surface contact. Data is pooled from all cells analyzed in data sets for (**b**, **c**). Statistical analysis using a one-way ANOVA followed by the Tukey post-test. S.E.M. is shown. For **b–d**, **g**, **h**, all the data are pooled from 3 biological replicates. All images are oriented with leader TVC to the left. *$p < 0.05$, **$p < 0.005$, ***$p < 0.0005$. L leader, T Trailer

Since both Ddr and Intβ1 can function as collagen receptors[31,32], we tested whether they interact to regulate cell-matrix adhesion by quantifying TVCs' contacts with the epidermis following Ddrdn and/or Intβ1dn expression. To detect interactions between the Ddr and Intβ1 pathways we selected a 25 μg dose that produced lower levels of detachment defects (Supplementary Figure 3) to combine the two conditions and assay for synergy between the transgenes. Co-expression of Ddrdn and Intβ1dn did not significantly aggravate the detachment phenotype (Fig. 2a–d). Overall, there was no significant increase in the percent TVC pairs with adhesion defects in the Ddrdn/Intβ1dn doubles compared to single perturbations, suggesting that Ddr and Intβ1 function in overlapping pathways to regulate TVC/epidermis adhesion. These data indicate that altering Integrin-β1 and Ddr functions cause similar phenotypes characterized by a loss of adhesion to the ventral epidermis, cell

rounding, and unstable collective polarity. We conclude that Ddr and Integrin-β1 promote cell-matrix adhesion at the contact with the ventral epidermis thus permitting directed migration.

To further characterize the functional relationships between the two collagen-receptors, we asked whether Intβ1 is required for Ddr activation. To assay localization and activation of full-length Ddr, we used the minimal *Foxf* TVC enhancer and expressed a 3xHA-tagged version of Ddr at minimal detectable levels to avoid non-specific localization. We performed immunohisto-chemisty (IHC) combined with a proximity ligation assay (PLA) using anti-HA and anti-phospho-Tyrosine antibodies to visualize the phosphorylated form of Ddr[39,41]. Full-length Ddr::3xHA localized to the ventral plasma membrane and intracellular vesicles (Fig. 2g). However, phosphorylated Ddr preferentially localized to the ventral TVC surface, which contacts the epidermis (Fig. 2g). Quantitative analysis showed that although percent of phosphorylated Ddr varied generally, ~12% of expressed Ddr was phosphorylated at any given time.

To test whether Ddr activation on the ventral/epidermal side of migrating TVCs requires integrin-based cell-matrix adhesion, we repeated the PLA assay in embryos expressing the B7.5 lineage-specific *Mesp > Intβ1^{dn}* transgene. We found that Intβ1^{dn} misexpression reduced Ddr phosphorylation levels to ~6% ($p < 0.005$, two-way ANOVA test followed by the Bonferroni post test), and the enrichment of activated Ddr on the ventral cell surface of TVCs was lost (Fig. 2g), indicating that Intβ1 activity is required to localize and activate Ddr on the ventral/epidermal side of migrating TVCs. These observations reinforce the notion that Ddr and Intβ1 interact to promote TVC adhesion to the extracellular matrix on the epidermal side of migrating cells.

**The endoderm contributes to adhesion and Ddr activation.** Previous work showed that expressing a dominant negative form of the small GTPase Sar1 (Sar1^{dn}) with the endoderm-specific *Nkx2–1* enhancer inhibits ER-to-Golgi transport and secretion from the endoderm, causing a tumbling TVC phenotype reminiscent of that observed with Ddr^{dn} and Intβ1^{dn} (Fig. 3a, b). We quantified contacts between the TVCs and the epidermis to determine if TVC cell-matrix adhesion was impaired in *Nkx2–1 > Sar1^{dn}* -expressing embryos similarly to Ddr^{dn}/Intβ1^{dn} misexpression. *Nkx2–1 > Sar1^{dn}* -mediated inhibition of secretion in the endoderm caused a significant reduction of TVC surface contact with the epidermis (Fig. 3c), suggesting secretion from the endoderm enables cell-matrix adhesion between the migrating TVCs and the ventral epidermis.

To test whether endodermal cues potentiate Ddr function in TVCs, we combined secretion inhibition in the endoderm with TVC-specific Ddr^{dn} misexpression and quantified contacts between TVCs and epidermis. We observed a significant enhancement of TVC detachment phenotype by combining *Nkx2–1 > Sar1^{dn}* and *Foxf > Ddr^{dn}* compared to either perturbation alone (Fig. 3c), suggesting that secreted endodermal cues and Ddr function in parallel pathways regulating TVC adhesion. To probe the functional interaction between the endoderm and Ddr, we tested if secretion from the endoderm is required for Ddr activation in migrating TVCs. Although we did not detect a statistically significant reduction in Ddr phosphorylation upon electroporation of *Nkx2–1 > Sar1^{dn}*, this could result from mosaic incorporation of the transgene in endodermal progenitors. For instance, the percentage of TVCs without detectable Ddr phosphorylation -which is less sensitive to experimental mosaicism- increased, consistent with the notion that an endodermal cue contributes to Ddr activity (Fig. 3d, e).

**Col9-a1 is required for Ddr activation and adhesion.** As Ddr is a collagen receptor, we hypothesized that potential cues secreted by the endoderm could include collagens. *Col9-a1* expression was reported in the endoderm of developing *Ciona* embryos where it contributes to the development of the intestine[42]. We confirmed that *Col9-a1* is expressed in the endoderm adjacent to the ventral epidermis prior to the onset of TVC migration (Fig. 4a), potentially acting as a source of extracellular collagen for subsequent TVC migration. *Col9-a1* is also expressed strongly in the notochord and endodermal strand that runs the length of the tail (Fig. 4a).

To test whether endoderm-derived Col9-a1 acts as an extracellular signal promoting Ddr activation and cell-matrix adhesion, we used CRISPR/Cas9 to mutagenize *Col9-a1* in early vegetal blastomeres using a *Foxd* enhancer[43–45] to express Cas9. We used sgRNAs targeting the *Col9-a1* locus (Supplementary Figure 5a, b), and validated the efficiency of CRISPR/Cas9-induced *Col9–1a* knockouts using RNA probes against *Col9-a1* transcripts in control (*EBF*) and *Col9-a1^{CRISPR}* backgrounds (Supplementary Figure 5c). We measured contacts between the TVCs and ventral epidermis following CRISPR/Cas9-induced *Col9-a1* mutagenesis in endoderm progenitors. *Col9-a1* gene inactivation in endoderm progenitors produced a significant TVC detachment from the ventral epidermis, which mimicked the Ddr^{dn}, Intβ1^{dn} and Sar1^{dn} phenotypes. This indicated that endodermal secretion of collagen Col9-a1 is required for TVC cell-matrix adhesion (Fig. 4b).

Since loss of Col9-a1 function phenocopies Ddr^{dn} misexpression, and Ddr presumably binds collagens, we tested whether endodermal Col9-a1 is required to activate Ddr in TVCs. We used CRISPR/Cas9 to mutagenize *Col9-a1* in the endoderm and quantified Ddr activation by PLA. Loss of Col9-a1 in the endoderm significantly reduced Ddr activation its ventral localization in TVCs (Fig. 4c). Therefore, Col9-a1 promotes Ddr activation either directly or via an indirect effect on ECM organization (see discussion).

As TVCs migrate into the trunk, they intercalate between the endoderm, which deforms and envelops their dorsal surface, and the ventral epidermis, which provides the substrate for TVC migration[13,16]. Since Col9-a1 originates from the endoderm, whereas Ddr is activated on the ventral side that contacts the epidermis, we sought to visualize the localization of Col9-a1 secreted from the endoderm. We generated a Col9-a1::GFP fusion expressed in the endoderm using the *Nkx2–1* enhancer (*Nkx2–1 > Col9-a1::GFP*). Col9-a1::GFP was secreted from the endoderm and accumulated in the extracellular matrix where the GFP signal was detectable between migrating TVCs and the ventral epidermis, whereas the TVC/endoderm interface was devoid of Col9-a1::GFP (Fig. 4d). Expression of Sar1^{dn} in the endoderm resulted in Col9-a1::GFP proteins accumulating in large vesicles inside endodermal cells and abolished its localization to the TVC-epidermis boundary (Fig. 4d).

These results suggest that blocking Col9-a1 secretion from the endoderm alters proper ECM formation on the basal side of the epidermis, and thus inhibits Ddr activation in TVCs and cell-matrix adhesion. To test this possibility, we attempted to rescue *Nkx2–1 > Sar1^{dn}*-induced de-adhesion phenotypes by providing Col9-a1::GFP from an alternative source. Expressing Col9-a1::GFP from the epidermis using an *EfnB* driver[13] restored the extracellular GFP signal in the matrix lining the ventral epidermis (Fig. 4d). We quantified contacts between TVCs and the ventral epidermis following misexpression of Sar1^{dn} in the endoderm, together with *EfnB > Col9-a1::GFP* or *EfnB > GFP* as control. Resupplying Col9-a1 to the epidermal ECM was sufficient to

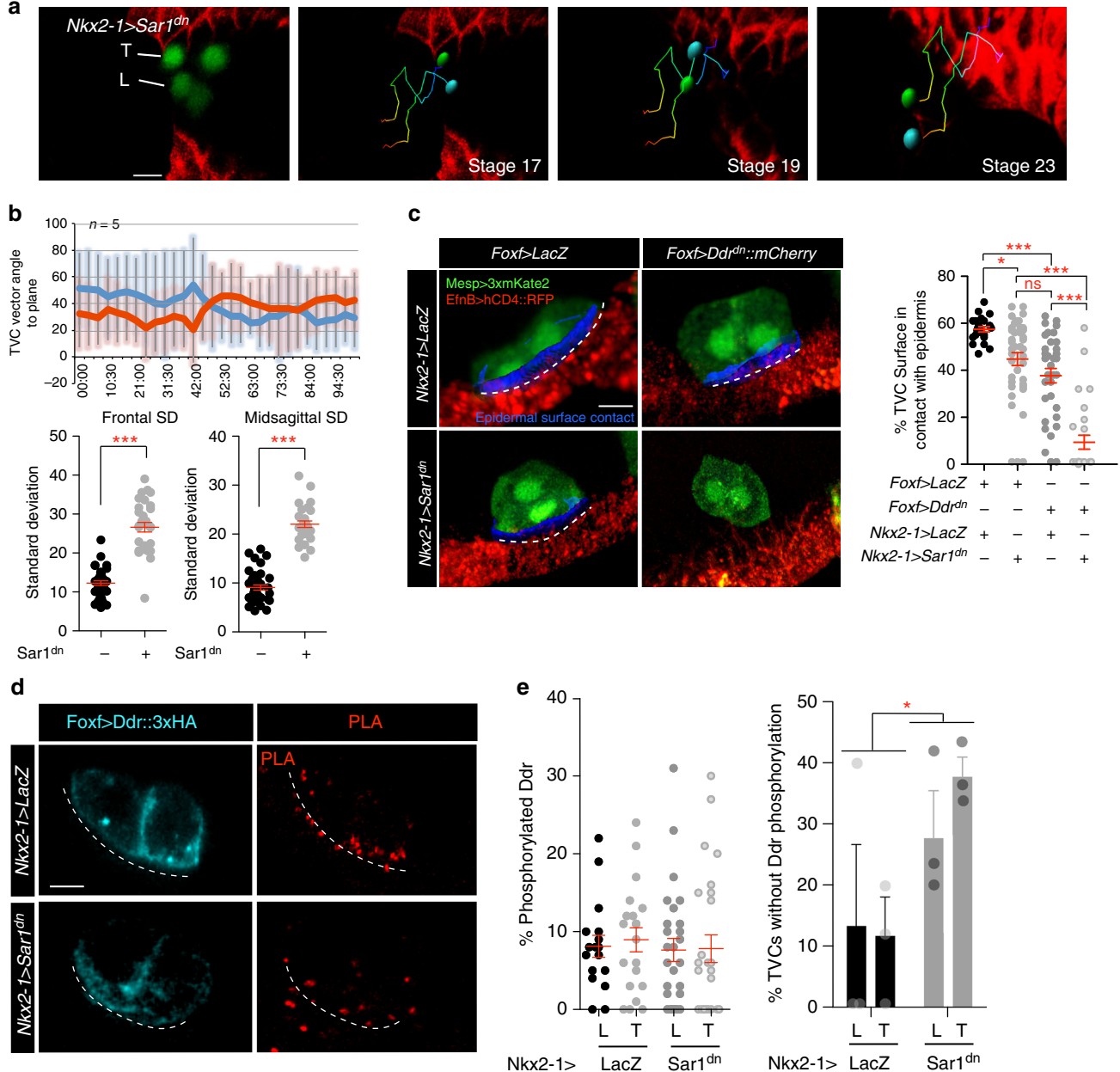

**Fig. 3** An endodermal cue potentiates TVC adhesion and Ddr activation. **a** Positional tracking of TVC migration in embryos expressing endodermal *Nkx2-1 > Sar1^dn* to block secretion. Tracks are time coded from early (blue) to late (red). Leader = cyan, trailer = green. Scale bar = 20um. **b** Quantitation of migration angles to frontal and sagittal planes is shown as averages at each time point and standard deviation from average. Statistical significance established using the Wilcoxon Rank Sum test. S.E.M is shown. **c** Percent of TVC-pair surface in contact with epidermis. The B7.5 lineage is marked with *Mesp > 3xmKate2*, epidermis is marked with *EfnB > hCD4::RFP*. Surface of contact is shown in blue and marked with a dashed line. Scatter plot shows average surface contact under each condition. Statistical analysis using one-way ANOVA followed by the Bonferroni post hoc test. S.E.M is shown. Scale bar = 10 μm. **d** PLA of *Foxf > Ddr::3xHA* under control (*Nkx2-1 > LacZ*) and *Nkx2-1 > Sar1^dn* conditions. Dotted line marks the ventral surface of TVCs. Scale bar = 10 μm. **e** Scatter plot on the left shows percent phosphorylated Ddr. S.E.M. is shown. Bar graph on the right shows the average percent of TVCs with no detectable Ddr phosphorylation. For **c** and **d** data are pooled from three biological replicates. All images are oriented with leader TVC to the left. *$p <$ 0.05, **$p < 0.005$, ***$p < 0.0005$. L leader, T Trailer

rescue the TVC adhesion phenotype induced by *Nkx2–1 > Sar1^dn* expression (Fig. 4e). This indicates that the endoderm contributes to TVC migration by depositing Col9-a1 onto the basal ECM of the epidermis, most likely prior to the migration of the TVCs, which rely on this cue to activate Ddr and adhere to their epidermal substrate.

**A Vegfr vs. Col9-a1/Ddr/Intβ1 antagonism modulates adhesion**. In principle, excessive cell-matrix adhesion would cause TVCs to flatten onto to the ventral epidermis. We occasionally observed this phenotype following TVC-specific misexpression of Vegfr^dn, which significantly reduced cell sphericity and produced a small increase in the average fraction of TVC surface in contact with

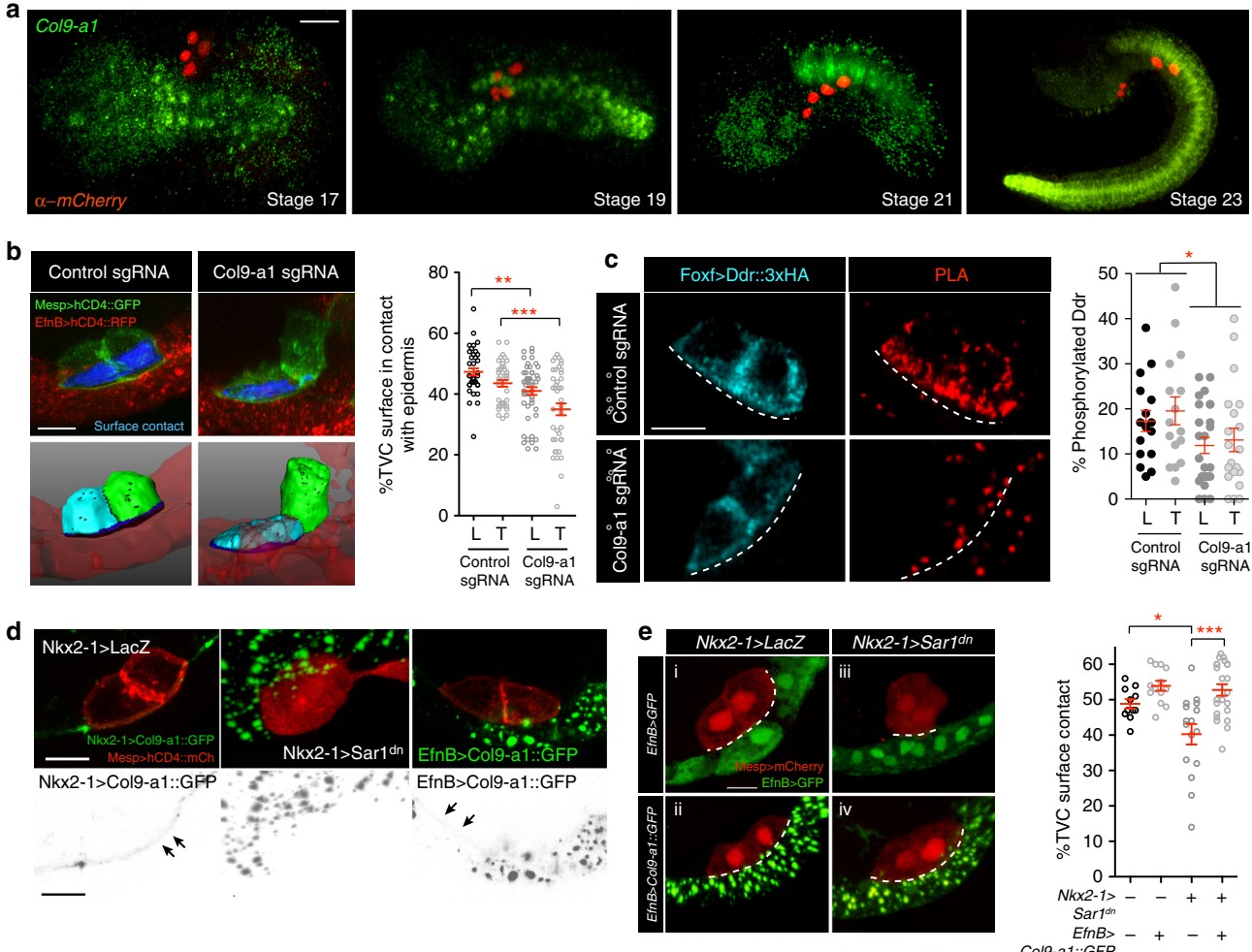

**Fig. 4** Col9-a1 is an endodermal cue that promotes TVC adhesion and Ddr activation. **a** Endogenous expression of *Col9-a1* in the developing *Ciona* embryo. Micrographs are oriented with anterior to the left, an in situ RNA probe is used to visualize *Col9-a1* transcripts. B7.5 lineage is visualized using *Mesp > H2B::mCherry*. Scale bar = 60 μm. **b** CRISPR mutagenesis of the *Col9-a1* locus in the endoderm using a *Foxd*-driven Cas9. Blue surfaces are sites of contact. Quantitation of TVC surface area in contact with epidermis adhesion is shown on the right. Statistical analysis using two-way ANOVA and the Bonferroni post hoc test. S.E.M. is shown. Scale bar = 10 μm. **c** PLA to detect phosphorylated full-length Ddr under control (EBF CRISPR) and CRISPR conditions targeting the *Col9-a1* locus for mutagenesis. Statistical analysis using two-way ANOVA. S.E.M. is shown. Scale bar = 10 μm. **d** Secretion and localization of an endodermally/epidermally expressed Col9A::GFP fusion. TVC membrane is visualized using *Mesp > hCD4::mCh*. First panel shows secretion and localization of Col9::GFP fusion. Middle panel shows block in endodermal secretion using Sar1dn. Gray scale panels show the corresponding inverted GFP channel. Arrows point to the accumulation of Col9-a1 in the extracellular matrix. Scale bar = 10 μm. **e** Rescue of *Nkx2-1 > Sar1dn* TVC adhesion defects with epidermally expressed *Col9-a1*. Epidermis is marked with *EfnB > GFP*. Note that in the lower panels *EfnB > GFP* is expressed but is less bright in comparison to the Col9-a1::GFP trafficking vesicles. Dotted line outlines the surface of contact. Quantitation of TVC adhesion is shown to the right of the micrographs. Scale bar = 10 μm. Statistical analysis using one-way ANOVA followed by the Bonferroni post hoc test. S.E.M. is shown. For **b**, **c** data are pooled from three biological replicates, **e** is pooled from two biological replicates. All images are oriented with leader TVC to the left. *$p < 0.05$, **$p < 0.005$, ***$p < 0.0005$. L leader, T Trailer

the epidermis (Fig. 5a–c). Since Vegfr$^{dn}$ expression altered TVC migration (Fig. 1e, g), and VEGF receptors and integrin complexes interact functionally in various contexts[46–48], we tested whether Vegfr signaling modulates Ddr/Intβ1-mediated TVC-matrix adhesion. Co-expression of Vegfr$^{dn}$ partially suppressed the de-adhesion phenotypes produced by Ddr$^{dn}$ (Fig. 5a, b), suggesting that a steady-state antagonism between baseline levels of Vegfr and Ddr signaling controls TVC adhesion to the epidermal matrix.

We tested whether components of Col9-a1/Ddr/Intβ1-mediated cell-matrix adhesion system modulate Vegfr activity using PLA assays to quantify Vegfr phosphorylation following CRISPR/Cas9-mediated mutagenesis of *Col9-a1* in endoderm progenitors, and misexpression of either Ddr$^{dn}$ or Intβ1$^{dn}$ in the

TVCs. In control TVCs, we observed most Vegfr::HA signal in cytoplasmic punctae, where phosphorylated Vegfr represented about 20% of the total pool on average (Fig. 5d–f). CRISPR mutagenesis of the *Col9a-1* locus resulted in an increase in the proportion of phosphorylated Vegfr to ~30% of the total pool with a preferential localization at the ventral membrane contacting the epidermal matrix. This suggests that extracellular Col9-a1 locally suppresses Vegfr activation at the ventral membrane (Fig. 5d). By contrast, Intβ1$^{dn}$ did not affect Vegfr phosphorylation levels or localization (Fig. 5f), whereas Ddr$^{dn}$ overexpression significantly decreased Vegfr phosphorylation (Fig. 5e). These results are consistent with the hypothesis that Col9-a1, Ddr, and Vegfr form an incoherent feedforward circuit that may modulate cell-matrix adhesion in migrating TVCs

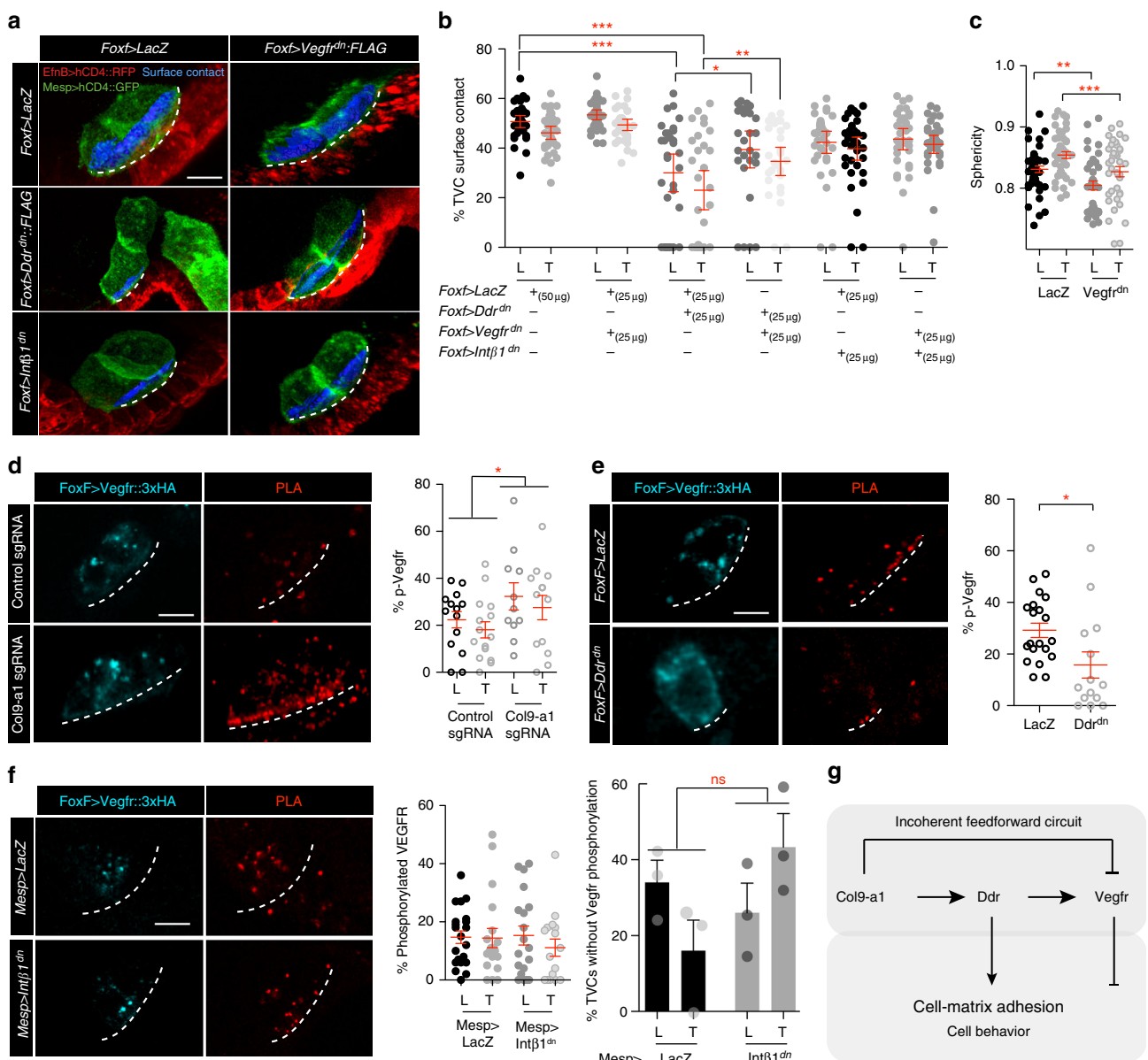

**Fig. 5** Vegfr modulates TVC adhesion through Col9-a1/Ddr/Intβ1 antagonism. **a, b** Regulation of TVC adhesion through opposing pathways. TVCs are visualized using *Mesp > hCD4::GFP*, epidermis is visualized using E*fnB > hCD4::RFP*. Extent of epidermal surface content is shown with dashed line. Blue surfaces represent portion of TVC in contact with epidermis. Scale bar = 10 μm. **b** Quantitation of TVC L/T surface in contact with epidermis under conditions that combine Vegfr[dn] and Ddr[dn] or Intβ1[dn]. Statistical analysis using two-way ANOVA followed by the Bonferroni post hoc test. 95% C.I. is shown. **c** Sphericity of leader/trailer TVCs under control and Vegfr[dn] conditions. Statistical analysis using two-way ANOVA. S.E.M. is shown. **d–f** PLA assay to detect levels of phosphorylated Vegfr under endodermal Col9a-1 CRISPR, Ddr[dn], and Intβ1[dn], conditions. Statistical analysis using two-way ANOVA. S.E.M. is shown. Scale bar = 10 μm. **g** Proposed model of an incoherent feed-forward loop regulating cell adhesion. For all graphs, data is pooled from three biological replicates. All images are oriented with leader TVC to the left. *p < 0.05, **p < 0.005, ***p < 0.0005. L = leader, T = Trailer

(Fig. 5g). The effects of Ddr[dn] suggest that Ddr may be required independently of its effects on cell-matrix adhesion to promote Vegfr signaling.

**Asymmetric Col9-a1/Ddr signaling establishes TVC polarity.** Previous studies showed that surrounding tissues contribute to canalizing TVC behavior towards collective leader/trailer polarity and directed migration[13]. The prospective leader cell initially contacts the endoderm, whereas the future trailer contacts the mesenchyme. This asymmetry suggests that an endodermal product, such as Col9-a1, contributes to establishing collective leader-trailer polarity prior to migration. To test if differential contact with the endoderm may determine asymmetric exposure to Col9-a1, we imaged Col9-a1::GFP deposition in the ECM and quantified colocalization with the TVC membranes prior to and shortly after onset of TVC migration (Fig. 6a). *Nkx2–1*-driven Col9-a1::GFP gradually accumulated in the ECM between stage 17 and 19, and clearly labeled the TVC/epidermis interface by stage 21, when TVCs begin their migration. Col9-a1::GFP foci initially colocalized more extensively with the prospective leader cell membrane, compared to the presumptive trailer (Fig. 6a).

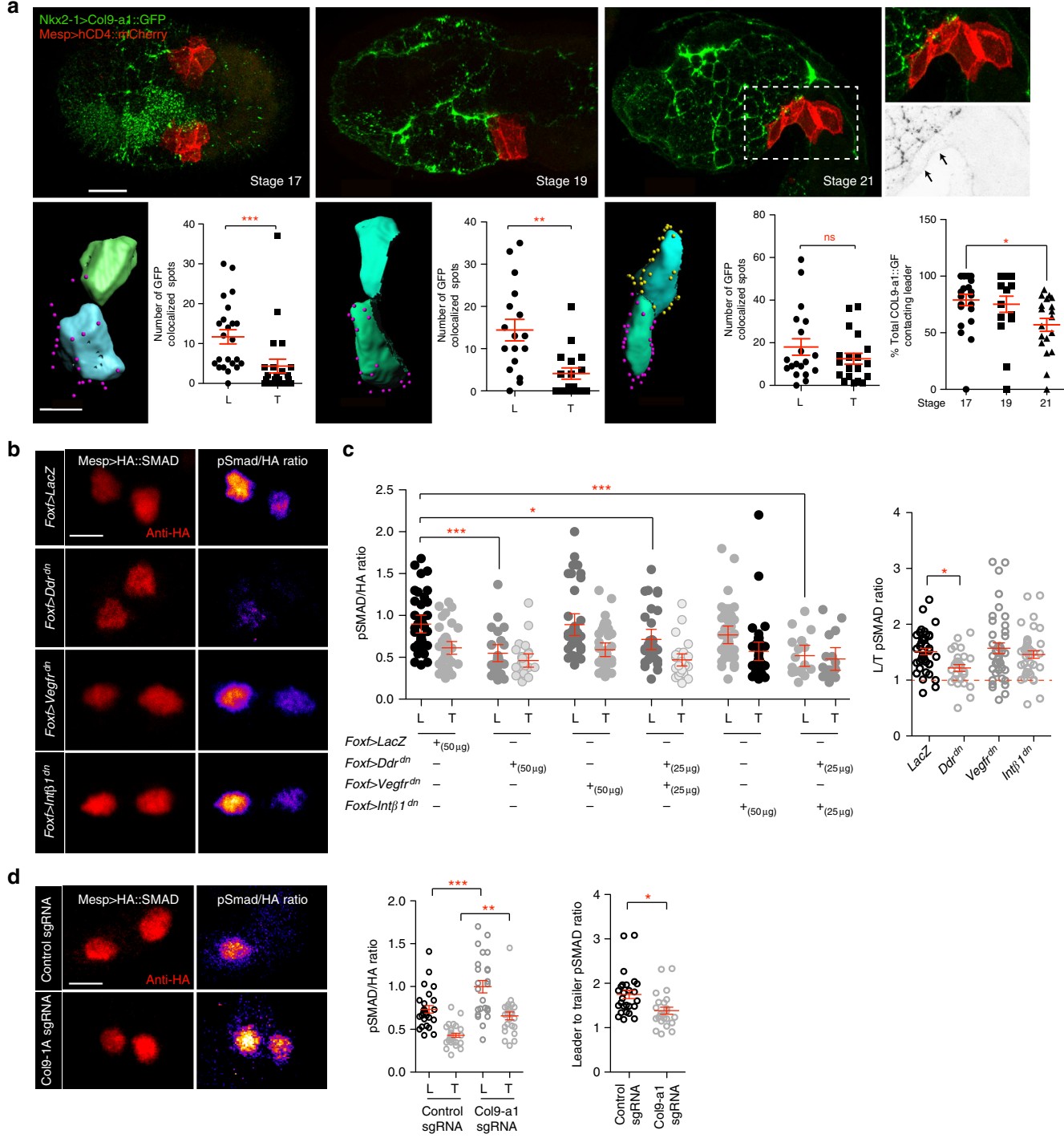

This suggests that at early developmental stages, asymmetric exposure to Col9a1::GFP provides a potential polarizing signal to the TVCs. As cells initiated migration, endoderm-derived Col9-a1::GFP colocalized similarly with leader and trailer. This implies that Col9-a1 induced polarity must be established prior to the onset of migration, and subsequently maintained through migration.

Collective TVC polarity is evidenced by morphological differences between the leader and trailer cells[13,16] (see also Figs. 1, 2), but no molecular marker of TVC polarity has been identified. We previously proposed that TVCs are exposed to varying levels of BMP-Smad signaling[49]. To test whether

BMP-Smad signaling is leader-trailer polarized in *Ciona* TVCs, we developed a biosensor by humanizing the C-terminus of *Ciona robusta* Smad1/5/8 and adding an HA tag at the N-terminus. Using both an anti-HA and an anti-phospho-SMAD5 antibody, which cross-reacts poorly with endogenous *Ciona* proteins, we quantified BMP-Smad signaling in the B7.5 lineage as the pSmad/HA ratio. This sensor responded to defined perturbations of the BMP-Smad pathway (Supplementary Figure 7), indicating that it serves as reliable readout for endogenous signaling. In control embryos, the pSmad/HA levels were ~1.75 times higher in leader compared to trailer cells (Fig. 6b, d). Therefore, we used changes in pSmad/HA levels to

**Fig. 6** Asymmetric exposure to Col9-a1 polarizes TVCs through an adhesion-independent function of Ddr. **a** Distribution of Col9-a1::GFP expressed in the endoderm at stages 17–21, prior to and at onset of TVC migration. Boxed region show localization of Col9-a1::GFP fusion ventral to migrating TVCs at stage 21. Arrows point to ventral accumulation of Col9-a1::GFP. Calculation of total exposure of leader or trailer to endodermal collagen at each embryonic stage. Presumptive leader = blue, presumptive trailer = green. Purple spheres indicate collagen spots proximal to the leader TVC, yellow spheres are proximal to the trailer TVC. Scatter plots show total number of collagen contact sites for leader or trailer at each given time point. Statistical analysis using the Wilcoxon Rank Sum test. Last scatter plot shows total number of Col9-a1::GFP spots contacting leader TVC over time. Statistical analysis using a one-way ANOVA followed by the Tukey post test. S.E.M. is shown. Whole embryo scale bar = 30 μm, rendered image scale bar = 15 μm. **b** Quantitation of pSmad activity under conditions perturbing TVC adhesion. Ratios between the pSmad and HA channels are calculated by dividing the intensity of the pSmad channel by the intensity of the HA channel. Ratio is color-coded using the Fire Look Up Table (LUT) with lighter colors indicating a higher ratio. Scale bar = 5 μm. **c** Normalized levels of pSmad in L/T under perturbation conditions and ratios of total leader:trailer pSmad levels. Statistical analysis using two-way ANOVA followed by the Bonferroni post hoc test and one-way ANOVA followed by the Tukey post hoc test. **d** pSMAD staining of control migrating TVCs and TVCs under endodermal Col9-a1 CRISPR mutagenesis conditions. L:T ratios and normalized L/T pSmad levels are shown to the right. Statistical analysis using two-way ANOVA followed by Bonferroni post hoc test for normalized pSMAD levels and Wilcoxon Rank Sum test for leader/trailer ratios. S.E.M. is shown. For a,d,data is pooled from two biological replicates, for c data is pooled from three biological replicates. Scale bar = 5μm. All images are oriented with leader TVC to the left. *$p < 0.05$, **$p < 0.005$, ***$p < 0.0005$. L leader, T Trailer

assay the roles of Col9–1, Ddr, Intβ1 and Vegfr signaling in establishing and maintaining collective leader/trailer polarity.

We quantified changes in BMP-Smad signaling following perturbations of Ddr, Intβ1, and Vegfr functions to test if adhesion to the ECM regulates polarity. TVC-specific expression of Ddr$^{dn}$ reduced the absolute PSmad/HA ratios and abolished differences between leader and trailer cells, suggesting that Ddr promotes BMP-Smad signaling and contributes to establishing and maintaining TVC polarity. By contrast, misexpression of neither Intβ1$^{dn}$ nor Vegfr$^{dn}$ altered pSmad/HA levels or asymmetry in TVCs (Fig. 6b, c). In an attempt to reveal cryptic activity, and because Vegfr appeared to antagonize the functions of Ddr in cell-matrix adhesion, we tested whether Vegfr$^{dn}$ or Intβ1$^{dn}$ could modulate the effects of Ddr$^{dn}$ on BMP-Smad signaling. We did not observe any significant difference in pSmad/HA levels in either of these conditions compared to Ddr$^{dn}$ alone. This suggests that Intβ1 and Vegfr act upon cell-matrix adhesion, whereas Ddr coordinates collective polarization upstream of adhesion by independently regulating each pathway.

As endoderm-derived Col9-a1 is necessary to activate Ddr in the TVCs, and newborn TVCs may be differentially exposed to extracellular Col9-a1, we probed BMP-Smad signaling following CRISPR/Cas9-induced *Col9-a1* mutagenesis. Unexpectedly, *Col9-a1* inactivation in the endoderm caused a general increase of pSmad/HA levels in both leader and trailer cells, and a slight reduction in leader-trailer ratio (Fig. 6d). Our results are consistent with a model whereby Col9-a1 regulates BMP-signaling via an incoherent feedforward signaling circuit (Fig. 7).

## Discussion

In embryos, the behaviors of progenitor cells emerge from integration and coordination of inputs from intrinsic factors, including transcriptional regulation of cellular effector genes, and biochemical and mechanical extrinsic determinants[1]. Here we showed that in migratory cardiopharyngeal progenitors of the tunicate *Ciona*, the transcription factor Foxf is necessary to upregulate the collagen receptor Ddr, which senses the extracellular matrix and functions to promote both cell-matrix adhesion and collective polarity through presumably independent, incoherent feed-forward pathways.

This study corroborates preceding transcriptional profiling results showing that Foxf, an early component of the cardiopharyngeal GRN, previously shown to impact protrusive activity by regulating the expression of the small GTPase Rhod/f[16], also connects the GRN to cellular effectors capable of sensing and responding to the extracellular milieu. This suggests that Foxf regulates a cell-matrix adhesion module, as previously

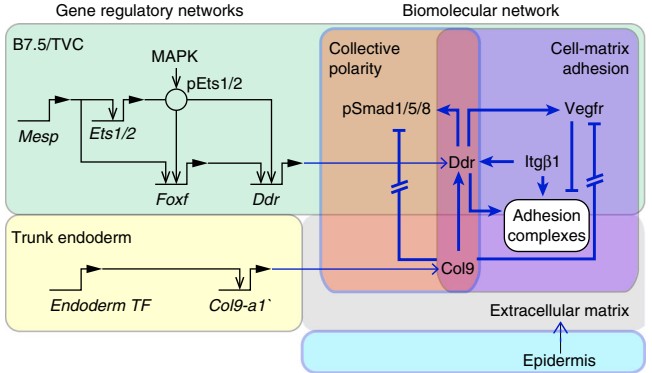

**Fig. 7** Integration of transcriptional and biomolecular networks to produce directional migration. Intersection of fate-specification through the cardiac GRN in the B7.5/TVC lineage (green) and downstream cellular effectors regulating ECM adhesion and polarity (purple and orange) regulates adhesion and detachment of TVCs to ensure directional migration. Black arrows show transcriptional regulation, blue arrows show activity regulation Gene regulatory connections are derived from[14–16,18,21]. Exposure to extracellular Col9-a1 activates Ddr while blocking Vegfr activity. Additionally, Intβ1 contributes to promotion of Ddr activity and adhesion. Solid blue lines show interactions confirmed in this report

proposed[16]. FoxF1 was shown to regulate Integrin-β3 expression and cell-matrix adhesion in the mouse[50], suggesting an ancient and conserved connection between Foxf homologs and the regulation of cell-matrix adhesion.

Previous studies used a dominant negative form of Foxf, with the WRPW repressor motif fused to the DNA-binding domain, to propose that Foxf activity controls TVC migration[15]. This reagent produced more severe migration phenotypes than our newly developed CRISPR/Cas9 assay, which nevertheless strongly inhibits *Foxf* expression (Gandhi et al., 2017[36] and Fig. 1b). This discrepancy suggests that the repressor form of Foxf caused non-specific defects; such as inhibiting targets of other Forkhead box family factors expressed in the B7.5 lineage[16], which may not be affected by CRISPR-induced Foxf loss-of-function. Future studies will clarify the functions of these other Fox factors at the transcription-migration interface.

While TVC-specific transcriptional inputs control the expression of cell-matrix adhesion determinants, asymmetric adhesion to the extracellular matrix was previously shown to polarize the founder cells and permit localized TVC induction, and thus *Foxf*

expression[51]. Precocious misexpression of Ddr$^{dn}$ using the *Mesp* driver inhibited this integrin-dependent TVC induction, thus supporting the notion that Ddr$^{dn}$ interferes with cell-matrix adhesion (Supplementary Figure 8). This illustrates the notion that cell fate specification and subcellular processes are dynamic and interdependent components of the transcription-cell behavior interface, which is best understood as an integrated system.

Our results indicate that at least two putative collagen receptors, Ddr and Intβ1, contribute to cell-matrix adhesion during TVC migration. Discoidin-domain receptors and integrins were shown to interact during cell adhesion to collagen matrices[34,52,53]. Whereas the function of integrins as cell-matrix adhesion receptors mechanically coupled to the cytoskeleton is well described[54,55], Ddr homologs were previously shown to bind collagens independently of integrins[39,53,56], and they appear to function as weak adhesion molecule and/or an ECM sensor, which can promote cell-matrix adhesion through other pathways, including integrins[34,57]. Our data suggest that Intβ1 is required in this context for robust activation of Ddr on the ventral TVC surface, although it is unclear if the primary role of Intβ1 is to localize Ddr to the ventral surface thereby allowing it to interact with the ventrally localized collagens.

One feature of our model is a functional antagonism between Vegfr signaling and Ddr-dependent cell-matrix adhesion. Although ligands and signal transduction pathways remain to be characterized, the antagonism appears to occur in part through local inhibition of Vegfr at the ventral membrane, where Ddr and Intβ1 bind the ECM (Fig. 5). While functional interactions between VEGF receptors and integrin complexes are largely context-dependent[46], VEGFR2 and αvβ3 integrins were found to directly interact through their cytoplasmic tails[58] and integrin activity can promote VEGFR2 degradation[47,48]. Conversely, VEGFR2 activity is linked to turnover of integrin-based focal adhesion[59]. It is possible that Vegfr can induce endocytosis of integrin complexes, thus weakening cells' adhesion to the basal lamina. In *Ciona* cardiopharyngeal progenitors, the Ddr/Vegfr antagonism contributes to positioning cells between the endoderm and the epidermis, a hallmark of the mesoderm in triploblastic animals.

The ECM-sensing properties of Ddr also contribute to the establishment of distinct leader and trailer states, and we propose that this results from differential exposure to extracellular collagen. This is reminiscent of the roles of ECM components in establishing of leader cell states in wound scratch assays in vitro[60], and in guiding endomesoderm migration during gastrulation[61]. Moreover, DDR1 can interact with PAR complexes to polarize the actin cytoskeleton in individual cells[62]. Future studies will uncover molecular pathways by which Ddr controls collective polarity, and the elusive cell-cell communication mechanisms that we must invoke, by analogy with other systems[63], to explain maintenance of leader-trailer polarity throughout migration.

We reported differential BMP-Smad signaling as a reliable read-out of polarized leader/trailer states. Ddr and Col9-a1 are required for polarized Smad activation, which appeared largely independent of Intβ1-mediated cell-matrix adhesion and Vegfr signaling. This indicates that Ddr acts as a dual function receptor, promoting cell-adhesion and transducing polarity information through independent pathways. Although Ddr is activated by exposure to Col9-a1 and promotes BMP-Smad signaling, loss of Col9-a1 function increased BMP-Smad signaling. To reconcile these seemingly inconsistent results, we hypothesize that BMP ligands are sequestered by extracellular collagen and inaccessible for signaling, as reported in the Drosophila ovary[64]. In this model, extracellular Col9-a1 would control BMP-Smad signaling through an incoherent feed-forward signaling circuit, whereby activation of Ddr-mediated signaling would compensate ligand sequestration in the extracellular matrix. Generally, incoherent feed-forward circuits have been proposed to act as accelerators for oscillating signals[65] and other studies have found functions for this signaling structure in sequestration of cell surface receptors and specification of endoderm and mesoderm lineages[66,67]. Future studies will test these possibilities and explore the signal transduction pathways connecting Ddr to Integrin, BMP-Smad and Vegfr signaling.

## Methods

**Electroporation and transgene expression**. *Ciona robusta* (formerly known as *Ciona intestinalis* type A) adults were purchased from M-Rep, San Diego, Ca. Gametes were isolated by dissection and separated into artificial sea water. Sperm was activated with 2.5 µl 1 M NaOH in 10 ml artificial sea water, poured over the eggs and allowed to fertilize for 6 minutes with gentle agitation. Fertilized eggs were dechorionated using a solution of artificial sea water, pronase, and sodium thioglycolate for up to 7 min. Eggs were washed three times in artificial sea water and pulsed in a BioRad electroporator at 50 V. More details about electroporation of Ciona can be found in references[68–70]. The amount of DNA electroporated varied from 10 µg to 90 µg. Animals were reared at 16 to 20 °C. For proximity ligation assays, embryos were fixed in cold 100% methanol for 10 min. For in situ hybridization experiments, embryos were fixed for 2 h in 4% MEM-PFA, dehydrated in an ethanol series and stored in 75% ethanol at −20 °C as described[68,70]. Embryos used for direct visualization of fusions were fixed in 4% MEM-FA for 30 minutes, cleared with an NH$_4$Cl solution, and imaged using a Leica SP8 X Confocal microscope.

**Molecular cloning**. Coding sequences of Ddr (KH.C9.371) Vegfr (KH.C14.345), Fgfr (KH.S742.2), (Egfr (KH.L22.45), and Col9-a1 (KH.C8.248) were amplified from mid-tailbud *Ciona* cDNA libraries. To generate dominant negative constructs, cytoplasmic kinase domains were removed by truncating the coding sequence after the transmembrane and extracellular domains. Primer lists are available in Supplementary Table 1. To subclone larger protein-coding fragments we used a multi-fragment assembly approach using the NEB InFusion homologous recombination kits (Takara, Cat #121416). Coding regions that were greater than 2 kb in length were subdivided into overlapping kernels and ligations of up to 4 kernels were performed.

**Live imaging and TVC tracking**. To generate 4D datasets, 4.5 hpf old embryos were mounted on glass bottom microwell petri dishes (MatTek, part# P35G-1.5–20-C) in artificial seawater. Plates were sealed by piping a border of a mix of Vaseline and 5% (v/v) mineral oil (Sigma, item #M841–100 ml) and covered with a 22 × 22 Fisherbrand Cover Glass (item # 12–541-B). Embryos were imaged on a Leica inverted SP8 X Confocal microscope every 3.5 min for 4–5 h. B7.5 lineage nuclei and epidermal cell membranes were visualized using *Mesp > H2B::GFP* and *EfnB > hCD4::mCherry*[13], respectively, and TVC migration was tracked using Bitplane Imaris Software Spots module.

To express TVC position during migration in 3D we subdivide the embryo into quadrants using two conceptual orthogonal planes that bisect the developing embryo. The position of the sagittal plane is determined by the midline of the epidermis visible using the *EfnB > hCD4::mCherry* transgene. The frontal plane is orthogonal to the mid-sagittal plane and passes through the nucleus of the anterior ATM and the future position of the palps, the most anterior point of the embryo. Imaris (Bitplane) is used to calculate a vector line based on position of GFP + TVC nuclei.

**Proximity ligation assay**. To visualize phosphorylated RTKs each receptor was subcloned into an expression vector driven by either the *Mesp* (early expression) or *Foxf(TVC)bpFog* (late expression) enhancer and tagged with 3x hemagglutinin (HA). Expression vectors were electroporated into fertilized *Ciona robusta* embryos, which were raised at room temperature to desired stages as indicated. Embryos were fixed by washing in 1 ml of 100% methanol chilled on ice for 10 min and stored in 100% methanol at −20 C until processed. Fisherbrand slides were coated with 0.1% w/v Poly-L-Lysine (Sigma P8920) two times and allowed to dry. Embryos were rehydrated in a 0.1% PBT (0.1% Tween-20 in PBS) MeOH series of 3:7, 1:1, 7:3, 100% PBT for 15 minutes and mounted on the Poly-L-Lysine coated slides. Slides were blocked in three drops of Duolink Blocking reagent for 30 min at 37 °C. Rabbit Anti-HA (Abcam ab9110) and mouse anti-pTyr (Millipore 4G10 Platinum anti-Phosphotyrosine, Sigma-Aldrich, Cat # 05–321 ×) were used to visualize phosphorylated RTKs and a rat anti-HA antibody (Roche 11–867–423–001) was used to visualize total amount of RTK expressed in the embryo. All three antibodies were used at a concentration of 1:500 diluted in Duolink Antibody Diluent. Slides were incubated with primary antibody for 1 h at 37 degrees and washed 2x in home made Duolink Buffer A (0.01 M Tris, 0.15 M NaCl and 0.05% Tween 20 pH 7.4) for 15 min each in a Coplin Jar. Secondary Duolink anti-mouse and anti-rabbit antibodies conjugated to a DNA probe were used as directed by the Duolink manual. AlexaFluor-555 anti-rat secondary

antibody (Life Technology #A21434) was added at a 1:1000 dilution to the mix of secondary antibodies to detect total amounts of HA. 100 µl total of secondary antibodies were used per slide. Slides were incubated for 1 h at 37 °C, washed two times in home-made Duolink Buffer A for 15 min each in Coplin jars. Ligation reactions were carried out as recommended in the Duolink manual, but extended to 1 h at 37 °C. Slides were washed again two times in Duolink Buffer A for 15 minutes each in Coplin jars. Polymerase reaction was carried out as recommended, and extended to 2 h at 37 °C. Slides were washed two times in home-made Duolink Buffer B (0.2 M Tris and 0.1 M NaCl pH 7.5) for 15 min each and one time in 0.01% Buffer B for 15 min. 15 µl Duolink Mounting Media was added to each slide, 22 × 22 Fisherbrand coverslips (Fisher Scientific Cat # 12–541B) were placed on top and the slides were sealed with clear nail polish.

**Fluorescent in situ hybridization – FISH-IHC**. RNA hybridization probes were generated from late tailbud derived cDNA by subcloning the coding regions of the genes of interest into the pCRII-TOPO dual-promoter cloning vector (Invitrogen, KH461020) and PCR was performed to confirm the orientation of the insert. Coding regions were amplified using M13 Forward and Reverse primers and the PCR product was used as a template to synthesize RNA probes labeled with either digoxigenin or fluorescein using the SP6 polymerase. Primers for probe generation are listed in Supplemental Table 1. Embryos were fixed in 4% PFA solution and dehydrated in an increasing ethanol series for storage. Embryos were rehydrated in a methanol:PBS series and permeabilized with a Proteinase K solution. Embryos were then incubated with a DIG or Fluo labeled probes overnight. Hybridized embryos were washed in a formamide series and incubated with a secondary antibody overnight at 4°. Embryos were washed in TNT and mounted in Prolong Gold. For more details please see the detailed protocols in[68,70].

**CRISPR/Cas9 - guide RNA design**. Single guide RNAs (sgRNAs) targeting the *Col9-a1* locus (KH.C8.248) were designed and tested essentially as described[36,44]. To test the efficiency of the sgRNAs, we first mutated the Col9-a1 locus using the ubiquitous *Ef1a > nls::Cas9::nls* and the *U6 > sgRNA* constructs. Cutting efficiency was calculated based on Sanger sequencing of the sgRNA target region as described[36]. Cutting efficiency of *Col9-a1* sgRNAs peaked at 0.8 and 1.1 for sgRNAs targeting the 1st and 29th exon, respectively. We targeted the *Col9-a1* locus by combining two high efficiency Col9-a1 sgRNAs (sequences of sgRNAs given in supplemental table 1), expected to generate large deletions. To target the *Col9-a1* locus in endoderm progenitors, we used the vegetal hemisphere enhancer from *Foxd* (Kubo et al, 2010) to drive nlns::Cas9::nls expression. 25 µg of Cas9-expressing vector and a total of 80 µg of sgRNA-expressing constructs were used for each experiment. As a control, sgRNAs targeting the late-expressed *Ebf* gene were used throughout the manuscript.

**Design and cloning of short hairpin RNAs (shRNAs)**. We designed the Ciona short hairpin microRNA (Ci-shmiR) cassette based on the primiR structure of Cirobu.mir-2213-a (See Hendrix et al. 2010). The Ci-shmiR cassette (aaagcggccg caaagctagcataatgaacttcgtggccgtcgatcgtttaaagggaggtagtgaggtacctctagt ggatcc [cgcggccgctaggttcgtttaatggtctaaaaatcaGagcgtttagt**GTTTG**gagaccgagagagggtctactaa aactgcgcttattatcttctacgaacctgtaagtggc] agatctggccgcactcgagtttgatgaattccagctgagcg) was cloned downstream of the *Mesp* enhancer. Hairpins were cloned into the Ci-shmiR cassette using BsaI. We design haripins targeting the coding region and evaluate shmiR efficacy using GFP fusion protein as described in Wang et al.2013. Hairpins that knocked down the reporter were further validated by in situ hybridization for the target gene.

**BMP-Smad biosensor, pSmad staining and quantification**. The Mesp > HA:: Smad1/5/8^Hs was built by replacing the C-terminus of *Ciona* Smad1/5/8 by that of human SMAD5, using standard cloning procedures. A monoclonal Rat anti-HA (Roche 11–867–423–001, 1:200) antibody was used to evaluate total levels of expressed protein and to normalize pSMAD levels across multiple embryos. Polyclonal Rabbit anti-phospho-SMAD5 (Cell Signaling, 9516 s, 1:250) was used to label phosphorylated HA::Smad1/5/8^Hs proteins, and anti-beta-galactosidase (Promega 23781 1:500) against the protein product of Mesp > LacZ was used to visualize the B7.5 lineage. To quantitate pSMAD levels as ratios, we use the anti-beta-galactosidase signal to identified TVC nuclei as spots in Bitplane Imaris. We then took the quantified the average fluorescence of anti-pSmad and anti-HA in the leader and trailer TVC. Ratios of pSmad to HA were calculated to normalize for variable transgene expression. The sensor's response to BMP-Smad signaling was validated using expression of published constitutively active BMP receptor and Noggin[49], an extracellular inhibitor (Supplementary Figure 6).

**Image acquisition**. All images were acquired using the Leica SP8 X WLL Confocal microscope using the 63x glycerol immersion lens, NA = 1.44. Z-stacks of fixed embryos were acquired at the system optimized Z-step, 512 × 512 resolution, 600 Hz, and bi-directional scanning. Multiple HyD detectors were used to capture images at various wavelengths.

**Morphometrics analysis and surface contact calculation**. The membrane marker Mesp > hCD4::GFP was used to segment the TVCs and derive morphometric measurements such as sphericity, area, and volume in Bitplane Imaris using the Cell function. Z-steps were normalized to achieve equal voxel size in X, Y, and Z planes. TVCs were then segmented and resulting cells were exported to separate surfaces. Another surface was created for the epidermis using the EfnB > hCD4:: tagRFP or mCherry marker. Distance transformation was performed on each TVC and the epidermal surface was then used to mask the distance from the epidermal surface to the distance-transformed surface of each TVC. A new surface representing the surface of contact between the TVC and the epidermis was created using the masked distances. To calculate the percent of the TVC surface in contact with the epidermis the area of the surface contact was divided by two and then divided by the total area of the TVC.

**Colocalization**. To identify spots of colocalization the CoLoc module of Bitplane Imaris was used to create a colocalization channel. The threshold of each image was set by the signal of interest, e.g. to find colocalization between the Mesp > hCD4:: mCherry and Nkx2–1 > Col9-a1::GFP the threshold was set to include the red channel. A separate channel was then created based on the colocalization and each area of that channel was transformed into a spot. We then calculated the number of spots close to the leader and trailer by identifying spots within 5 microns of the TVC surface.

**Statistical analysis and data representation**. For Fig. 1g, we represent the straightness of the migration path by plotting total displacement against total track length and calculating the slope of the best fit line forced to intercept the graph origin. For all the data comparing two samples of continuous variables the Wilcoxon Rank Sum test (also known as the Mann–Whitney test) is used. For the data sets containing more than two conditions and taking into account cell type (Leader/Trailer) a two-way ANOVA followed by the Bonferroni post test is used. For all data sets containing nominal variables a chi-square test is used. $P$ values as reported as follows: $*p < 0.05$, $**p < 0.005$, $***p < 0.0005$.

**Reporting Summary**. Further information on experimental design is available in the Nature Research Reporting Summary linked to this article.

## Data availability
The full data supporting this article are available from the corresponding author upon reasonable request.

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

## Acknowledgements

We thank Justin Le Lorier for cloning the HA::Smad1/5/8$^{Hs}$ sensor. This work was supported by NIH F32 GM108369–01A1 post-doctoral fellowship to Y.Y.B., NIH F32 GM105216–01A1 post-doctoral fellowship to S.E.G., and NIH/NIGMS R01 GM096032 award to L.C.

## Author contributions

Y.Y.B. and L.C. designed the experiments, Y.Y.B., S.E.G., S.B. and W.W. performed the experiments. Y.Y.B. and L.C. wrote the paper.

## Additional information

**Competing interests:** The authors declare no competing interests.

