## [Peer Review File · Nature Communications]

Reviewers' comments:

Reviewer #1 (Remarks to the Author):

The paper by Bernadskaya et al uses the simple invertebrate chordate *Ciona* to examine the directed migration of pairs of bi-potential cardiopharyngeal progenitors (TVCs). As they migrate, TVCs organize into a leader and trailer and these cells have previously been well characterized. Here the authors focus on the role of RTKs, which were identified in a previous transcriptomics screen, in particular Ddr and Vegf receptors. Ddr and Vegfr are expressed in the earliest TVCs and expression is reduced in absence of FoxF function.

They use quantitative analysis of cell movements in live embryos using fluorescent markers for the epidermis and the B7.5 lineage. Time lapse recordings were done from very few embryos for each condition, the behavior is variable and after expression of dn mutants cell behavior becomes more variable. Overall, dnDdr expression resulted in cells failing to maintain relative leader/trailer position, whereas dnVegfr expression increased track straightness, but the effects seem marginal. (Fig 1)

The dnDdr phenotype was further characterized and leader cells become more spherical and have reduced contact with the epidermis. This phenotype was mimicked by altering the function of Integrin b1. However, the conclusion that Intb1 activity is required to localize and activate Ddr is not convincingly supported by the marginal effects observed. (Fig 2)

Inhibition of endoderm secretion significantly enhances the detachment phenotype, this data looks clearer, although still quite variable and there are fewer data points for dnDdr plus dnSar1 compared to the other combinations. (Fig 3)

They then use CRISPR/Cas9 system to 'inhibit' or 'mutate' Col9a1 and then examine the detachment phenotype, which seems to be more pronounced in the trailer but there is no significant difference in the leader cell. This is not mentioned in the text. There is no control shown for the efficiency of the gene editing and the degree of mosaicism or any off target effects. The conclusion that "Col9a1 is likely the collagen ligand responsible for Ddr activation in the TVCs" (lines 290, 291) is not supported by the data shown in Figure 4C. There is no significant difference in the % phosphorylated Ddr in presence of sgRNA for Col9a1. Furthermore, the logic underlying the ectopic Col9a1 expression experiment is not clear. Only because dnSar1 blocks secretion of overexpressed Col9a1 does not mean this is what normally occurs.

The proposed antagonism by Vegfr is not very convincingly shown (Fig 5A-C). Effects are marginal and it would be important to know the effect of 25ug of dnDdr on its own and 25ug dnIntgb on its own (use empty vector to make up same amount of total DNA). The data shown does not support the statement (line 328) "Co-expression of dnVegfr partially suppressed the de-adhesion phenotypes....". Thus, this "prominent feature" of their model (line 457) is poorly supported. The interpretation of the unexpected effects of dnDdr is also confusing. In the final section on the role of BMP-Smad signalling for polarization, the interpretation of the findings is very speculative. This is also repetitive with the final section of the discussion. The text referring to Fig 6: Line 392-394, "we tested whether dnVegfr etc....In neither case did we observe....etc." In Fig. 6C there is no data that shows that dnIntgb3 has no effect on dnDdr on BMP-Smad signalling.

In conclusion, the major claims of the paper are not convincingly supported. Although quantitative methods are applied, many effects are marginal. The speculative interpretation should be removed from 'results'.

Other issues:

Line 338, the proposed increase in phosphorylated Vegfr in presence of sgCol9a1 or dnIntgb1 is also not clearly supported by the data shown in Fig 5c

Line 213-215, "Co-expression of sub-optimal doses.....aggravated the detachment phenotype....etc." It is not clear what they mean here. Variability is high and results are essentially similar to single dn receptor experiments, 'aggravated' compared to what?

Line 226,226, should read Figure 2F

Line 227, "12-20% of tagged Ddr proteins were phosphorylated", need to refer to the data I cannot see that in Figure 2

Line 228, "...suggesting that there is a large pool of inactive Ddr proteins within the cells." It is not clear how this conclusion can be made based on overexpression of tagged proteins. The authors should remove this statement.

Line 234, should read Figure 2F, G

Line 254, should read Figure 3B, could not find a reference to Figure 3A

Line 265, that that

Line 445, dynamical (is not a word)

Reviewer #2 (Remarks to the Author):

The cardiac lineage of Ciona embryos represents the simplest example of collective cell migration with a pair of cardiopharyngeal progenitors (TVCs) undergoing a polarised migration following a well-characterised temporal sequence. In this study, the authors deploy an impressive panel of technically-challenging approaches to the Ciona system and analyse quantitatively roles of Discoidin domain receptor (Ddr) and cell-matrix adhesion during the collective migration of TVCs.

The major claims of this study are as follows. 1) The TVC-specific Ddr, in cooperation with Integrin-b1, promotes cell-matrix adhesion to epidermis. At reduced levels of cell-matrix adhesion, the directionality of migrating TVCs is severely affected. 2) An endodermally-derived collagen, Col9-a1, is deposited in the matrix lining of the ventral epidermis and activates Ddr at the ventral membrane of migrating TVCs. 3) BMP-Smad signalling is differentially activated between the leading and tailing TVCs and Ddr promotes BMP-Smad signalling independently of its role in cell-matrix adhesion. Finally, 4) Ddr promotes Vegfr signalling in TVCs independently of its role in cell-matrix adhesion. In contrast, cell-matrix adhesion and Vegfr signalling form an antagonistic relationship in TVCs. In order to highlight these findings succinctly, the authors integrate them in a simple schematic, which is presented as Figure 7.

As highlighted above, the current study involves major technical achievements. The successful application of Proximity Ligation Assay (PLA) to visualise a phosphorylated form of Ddr with subcellular resolution is impressive. I am also impressed with the cleverly-defined morphometric parameters in order to describe TVC migration quantitatively. The Christean lab also appears to have fully mastered the use of CRISPR/Cas9-mediated gene knockout in Ciona system.

My major concern of this study is that some of the major claims are not statistically supported (i.e. with p-values). These claims are as follows: Vegfr signalling negatively controls cell-matrix adhesion (Fig 5B), Cell-matrix adhesion negatively controls Vegfr signalling (Fig 5E,F), Integrin- β 1 promotes Ddr activation (Fig 2G), Col9-a1 promotes Ddr activation (Fig 4C). Many of these claims are based on PLA experiments. It is also important that the authors describe in detail how they quantified the PLA results and expressed them as "% Phosphorylated Ddr/Vegfr/RTK". I am wondering whether the results might be better represented by counting the number of PLA spots in the ventral and dorsal membranes and dividing it by total number of spots. Similarly, there is no statistical analysis for Fig 1E,F and Fig 3D despite conclusions being drawn. I suggest that the authors include statistical support for all experiments where possible or clearly state when conclusions are hypothetical or why statistical analysis was not possible. Otherwise, it seems odd to see statistical analysis on some data but not on others.

Minor points:

- 1) page 6/line 153, change Figure 1G to Figure 1E.
- 2) page 8/line 225, change Figure 2E to Figure 2F.
- 3) page 9/line 234, change Figure 2E,F to Figure 2F,G.

Reviewer #3 (Remarks to the Author):

This is an interesting report in which the authors investigated the mechanisms of collective polarity using migratory cardiopharyngeal progenitors of the tunicate *Ciona* as a model. Cardiopharyngeal progenitors (also termed as trunk ventral cells, TVCs) that polarize as leader and trailer cells, were found to express Discoidin domain receptor (Ddr) in a manner dependent on the transcription factor Foxf, a regulator of TVC migration. Ddr was found to co-operate with β 1-Integrin in TVCs, promoting cell adhesion to the epidermis, during TVC migration between the trunk endoderm and ventral epidermis. Endoderm-derived collagen 9-a1 (Col9-a1) was deposited into the basal epidermis and was found necessary for activation of Ddr at the TVC-epidermis interface. Vegfr was found to antagonize Ddr/ β 1-Integrin-mediated TVC adhesion to the ventral epidermis. Furthermore, Ddr regulated leader-trailer-polarized BMP-Smad signaling independently of its role in cell-matrix adhesion. Collectively, the authors propose a model where Ddr, acting downstream of cardiopharyngeal-specific transcription factors, regulates both collective polarity and directed TVC migration/adhesion.

Major comments:

1. The authors should report which statistical tests were used, as well as the number of independent experiments/embryos (n) in each figure. Since much of the data is based on quantitative image analysis, details on these should be added (e.g. is each data point derived from a different embryo).
2. The authors suggest that BMP signalling is enhanced in the Col9-a1 mutants (loss-of-function by CRISPR/Cas9), due to increased bioavailability of BMPs. Does Col9-a1 (type IX collagen) bind BMPs? Col9-a1 is very different from collagen IV that is known to bind BMP2/4, thus, it may not be plausible to assume that they function in a similar manner.
3. The authors conclude that Col9-a1 is the ligand for Ddr, since Col9-a1 deletion decreased Ddr phosphorylation (Fig.4). However, have the authors examined if loss of Col9-a1 affects deposition, organization or stability of other matrix components, thereby excluding potential indirect effects due to Col9-a1 loss that might prevent Ddr activation in their model? As suggested for this class of non-fibrillar collagens, type IX collagen is expected to link type II collagen fibers and the proteoglycans, thus contributing to the overall matrix organization in the cartilage. Which collagens activate Ddr in *Ciona*, are these still expressed in the mutant embryos and does the ligand specificity differ between *Ciona* and human Ddr? Col9-a1 staining is necessary to show the decrease in Col9-a1 expression in

mutants vs wild type embryos.

4. Fig. 2F, 3C and 4C show an almost complete loss of phospho-Ddr, whereas the quantifications in 2G, 3D and 4C are less striking, and lack significance – why? The authors should consider referring those results as a “trend”, and modify the figure titles in Fig 3 and Fig 4 accordingly.

5. The asymmetric exposure of TVCs to the endodermal derived Col9-a1 was concluded to stimulate leader/trailer polarization via BMP-Smad signalling (Fig. 6). Was polarization of TVCs affected/rescued in *Nkx2-1>dnSar1* embryos with epidermal expression of *EfnB>Col9-a1::GFP*?

6. In fig. 5, *dnVegfr* was found to partially suppress the de-adhesion phenotypes produced by *dnDdr* or *dnInt β1* in the TVCs, and *Col9-a1* inhibition and *dnInt β1* misexpression resulted in increased of *Vegfr* activity (Figure 5), suggesting that cell adhesion via *β1-Integrin* negatively regulates *Vegfr* activity. However, since *dnDdr* decreased *Vegfr* activity, other factors must be at play. Potential candidates affecting *Vegfr* activity would include ligand or phosphatase-mediated regulation of *Vegfr*; have the authors considered any of these and can these be discussed?

7. Determining the α -integrin subunit that pairs with $\beta 1$ -integrin would strengthen the authors' conclusions about the importance of *Col9-a1*.

Minor comments:

8. Fig 3A and Fig S5 were not referred to in the text.

9. Results in Fig S3, demonstrating the effects of *Ddr*, *Vegfr* and *β1-Integrin* on leader-trailer cell surface contact area are not discussed in the text.

10. Details of the *EfnB>Col9-a1::GFP* construct are missing in the methods.

11. Color coding in Fig S6 is missing.

First, we would like to thank you and the reviewers for their thoughtful comments. In general, and in light of the comments and suggestions, we have focused the revisions on consolidating the study, both experimentally and analytically, by repeating experiments and performing more in-depth analyses. As a complement, we have reserved the more speculative conclusions to the discussion (e.g. a possible role for collagen in sequestering BMP ligands in the ECM). Our statistical analyses were a major concern of all reviewers. We have addressed this concern by reviewing our statistics approach and now use the correct statistical tests to indicate significant differences between experimental conditions in our manuscript as well as being as clear as possible about the type of statistical analysis used. Below are detailed list of key modifications to the figures and responses to each comment.

Major changes:

Figure 1b and 1c: we added standard error calculations and used Chi-square test to estimate statistical significance.

Figure 1f: We added analyses of total displacement, which show that Ddr^{dn} misexpressing cells do not migrate as far.

Figure 1g: we added a panel showing the relationship between displacement and track length with slope of line indicating straightness of traveled path.

Figure 2b: We replaced sphericity with data derived from same experiment as the epistasis analysis to show that perturbation of either adhesion regulators increase sphericity, which is consistent with sphericity being anti-correlated with cell-matrix adhesion.

Figure 2c and 2d: we repeated epistasis/molecular interaction experiment between Ddr and Integrin-beta1 with low dose of Ddr^{dn} and Intβ1^{dn}, which we determined through a dose-response analysis, and scored detachment phenotype. This showed no enhancement of the detachment phenotype when conditions are combined, indicating that the two receptors operate in the same cell-matrix adhesion process.

Figure 2g: we performed a two-way ANOVA followed by the Bonferroni posttest, which showed statistically significant drop in Ddr activation.

Figure 2h: we added overall sphericity data binned by percent surface contact to demonstrate the relationship (i.e. anti-correlation) between adhesion and cell morphology.

Figure 3e: We repeated the Ddr PLA experiments following expression of Sar1^{dn} in the endoderm and used a two-way ANOVA followed by the Bonferroni posttest to test for statistically significant difference. We also calculated the percentage of TVCs lacking PLA signal in the three biological replicates and used a two-way ANOVA followed by the Bonferroni post test to check for statistically significant differences.

Figure 4c: we also used two-way ANOVA followed by the Bonferroni posttest to check for statistical significance.

Figure 5b: we replaced the previous epistasis panel with data using 25µg of dominant-negative-expressing construct per experiment. We also used two-way ANOVA followed by the Bonferroni posttest to check for statistical significance.

Figure 5d, 5e, 5f: We added statistical analysis using two-way ANOVA and Bonferroni posttest. In 5f calculated percent of TVCs that have no PLA signal in the three biological replicates and used a two-way ANOVA followed by the Bonferroni posttest to check for statistically significant differences.

Figure 5g: We added a model of the incoherent feedforward loop regulating adhesion.

Figure 6a: We added 1 more biological replicate.

RESPONSES TO REVIEWERS

Reviewer #1

The paper by Bernadskaya et al uses the simple invertebrate chordate *Ciona* to examine the directed migration of pairs of bi-potential cardiopharyngeal progenitors (TVCs). As they migrate, TVCs organize into a leader and trailer and these cells have previously been well characterized. Here the authors focus on the role of RTKs, which were identified in a previous transcriptomics screen, in particular *Ddr* and *Vegf* receptors. *Ddr* and *Vegfr* are expressed in the earliest TVCs and expression is reduced in absence of *FoxF* function.

They use quantitative analysis of cell movements in live embryos using fluorescent markers for the epidermis and the B7.5 lineage. Time lapse recordings were done from very few embryos for each condition, the behavior is variable and after expression of *dn* mutants cell behavior becomes more variable.

Response:

This limitation is inherent to our current methods to acquire 4D data sets, with only a fraction being usable (embryos move out of the field, etc...). While we are working on fixing these issues, we've used these datasets as entry points to characterize defined perturbations. To circumvent the low n number we've harnessed the fact that there are multiple time points acquired in each dataset to recover statistical power and estimate the impact of perturbation on parameter variability. As a general principle, this effect of molecular perturbations is a rather common outcome, hinting at the genetic underpinning of morphogenetic canalization. We have clarified this in the text.

Overall, *dnDdr* expression resulted in cells failing to maintain relative leader/trailer position, whereas *dnVegfr* expression increased track straightness, but the effects seem marginal. (Fig 1)

Response:

We have re-calculated and re-plotted the migration data in figure 1g to better illustrate the relationship between the track length and total displacement. The slope of the regression line reflects the straightness of the path traveled by the TVCs under each condition. It is now easier to compare migration paths. The plot illustrates the more convoluted path taken by *dnDdr*-expressing cells. Here too, these metrics are entry points to characterize the perturbations quantitatively, but we would caution against an over-interpretation of these data. As we explain in the paper, we think that these altered migration trajectories result from defects in cell-matrix adhesion, which we then focus on.

The dnDdr phenotype was further characterized and leader cells become more spherical and have reduced contact with the epidermis. This phenotype was mimicked by altering the function of Integrin b1.

However, the conclusion that Intb1 activity is required to localize and activate Ddr is not convincingly supported by the marginal effects observed. (Fig 2G)

Response:

We have used statistical analyses better adapted to the structure of the data. Namely, we use a two-way ANOVA with a Bonferroni post hoc test to identify statistically significant changes in levels of phosphorylated Ddr in two conditions (control vs dnIntegrin) and two cell states (leader vs trailer). We see a significant decrease in Ddr phosphorylation when dnIntb1 is expressed in the TVCs. Figure 2g is now Figure 2h.

Inhibition of endoderm secretion significantly enhances the detachment phenotype, this data looks clearer, although still quite variable and there are fewer data points for dnDdr plus dnSar1 compared to the other combinations. (Fig 3)

Response:

In these experiments, technical variability is inherent to the electroporation technique used in the field. While we're working on developing stable knock-in techniques, these are not available at the moment. As much as possible, we use appropriate negative controls (here misexpression of *LacZ*) and we've now performed a dose-response analysis for dnDdr and dn-Integrin-beta1 (Figure S3), to better characterize the specific effects of our perturbations.

We have increased the number of cells analyzed for Ddr phosphorylation under endodermal dnSar1 condition. While the results are still variable, this may be attributed to the mosaic inheritance of the dnSar1-expressing plasmid, which was difficult to account for (because there are already 3 different labels involved in this experiment). We therefore added a graph showing an increase in the number of TVCs without any Ddr phosphorylation and find that that proportion increases in a statistically significant manner as determined by a two-way ANOVA test when signaling from the endoderm is blocked. This result is now shown in Figure 3e.

They then use CRISPR/Cas9 system to 'inhibit' or 'mutate' Col9a1 and then examine the detachment phenotype, which seems to be more pronounced in the trailer but there is no significant difference in the leader cell. This is not mentioned in the text.

Response:

This is a good observation. In general, we sometimes see the trailer being more affected than the leader in terms of adhesion. However, this is mostly observed when dealing with fixed samples, which represent a single time point during TVC migration. As we generally fix these cells later in development, we cannot rule out that the apparent trailer did not start out as a lead that lost adhesion and was displaced by the trailer, which had retained some adhesion. We therefore do not probe deeper into this relationship at this time.

There is no control shown for the efficiency of the gene editing and the degree of mosaicism or any off target effects.

Response:

We have added supplementary figure 5 showing validation of the efficiency of the sgCol9-1a CRISPR using the analysis described in our protocol paper (Gandhi et al., Dev Biol (2017)).

The conclusion that “Col9a1 is likely the collagen ligand responsible for Ddr activation in the TVCs” (lines 290, 291) is not supported by the data shown in Figure 4C. There is no significant difference in the % phosphorylated Ddr in presence of sgRNA for Col9a1.

Response:

We re-analyzed the Col9-a1 CRISPR knockout effect on Ddr phosphorylation used a more adapted 2-WAY ANOVA and found a statistically significant decrease in Ddr phosphorylation. This information has been added to the figure. But this comment raises the important point that we cannot unequivocally demonstrate that Col9-a1 is the ligand for Ddr, which was also pointed by reviewer #3. In this regard, we have amended the text and summary model to leave open the possibility that Col9-a1 indirectly controls Ddr activity by impacting ECM organization.

Furthermore, the logic underlying the ectopic Col9a1 expression experiment is not clear. Only because dnSar1 blocks secretion of overexpressed Col9a1 does not mean this is what normally occurs.

Response:

We have clarified this part of the text. The effect of dnSar1 was shown as a proof of principle that dnSar can blocks secretion of Col9-a1 from the endoderm. The key point in the experiment is that expressing Col9-a1::GFP from the epidermis can rescue inhibition of secretion from the endoderm (current Fig 4e). This demonstrates that Col9-a1 is a key component missing from the ECM when secretion is inhibited in the endoderm.

The proposed antagonism by Vegfr is not very convincingly shown (Fig 5A-C). Effects are marginal and it would be important to know the effect of 25ug of dnDdr on its own and 25ug dnIntgb on its own (use empty vector to make up same amount of total DNA). The data shown does not support the statement (line 328) “Co-expression of dnVegfr partially suppressed the de-adhesion phenotypes....”. Thus, this “prominent feature” of their model (line 457) is poorly supported.

Response:

We repeated the epistasis assays with corrected amounts of plasmid DNA and observed a clearer rescue of the adhesion phenotype produced by dnDdr when dnVegfr was co-expressed. The impact of dnVegfr on cell sphericity (now presented in Fig 5C) is also consistent with a negative effect of Vegfr signaling in cell-matrix adhesion (i.e. cells spread out more onto the epidermal substrate when Vegfr is inhibited).

The interpretation of the unexpected effects of dnDdr is also confusing.

Response:

We have clarified the language of this interpretation. Namely, the effect of Ddr on Vegfr activity could be interpreted as independent from its effect on cell-matrix adhesion. For instance, this is consistent with Ddr's role regulating Bmp-Smad signaling and polarity.

In the final section on the role of BMP-Smad signalling for polarization, the interpretation of the findings is very speculative. This is also repetitive with the final section of the discussion. The text referring to Fig 6: Line 392-394, “we tested whether dnVegfr etc....In neither case did we observe....etc.” In Fig. 6C there is no data that shows that dnIntgb3 has no effect on dnDdr on BMP-Smad signalling.

Response:

We have removed discussion points from the results section and have corrected the Intb3 typo to read Int□1. More to the reviewer's point, we've consolidated the quantification and statistics in Figure 6b, c and still fail to detect an effect of Intgb1^{dn} or Vegfr^{dn} on BMP-Smad activity, whereas Ddr^{dn} and Col9-a1^{CRISPR} have a significant effect on BMP-Smad signaling.

In conclusion, the major claims of the paper are not convincingly supported. Although quantitative methods are applied, many effects are marginal. The speculative interpretation should be removed from 'results'

Response:

The more speculative conclusions have been moved to the discussion and the conclusions are now supported with consolidated data and appropriate statistical analyses.

Other issues:

Line 338, the proposed increase in phosphorylated Vegfr in presence of sgCol9a1 or dnIntgb1 is also not clearly supported by the data shown in Fig 5c

Response:

We have split figure 5c into 5d-e, Statistical analysis using two-way ANOVA and the Bonferroni post hoc test shows that there is a significant increase in the levels of phosphorylated Vegfr under Col9-a1 CRISPR knockout conditions. We have clarified that disruption of Int□1 does not have an affect on Vegfr phosphorylation.

Line 213-215, "Co-expression of sub-optimal doses.....aggravated the detachment phenotype....etc." It is not clear what they mean here. Variability is high and results are essentially similar to single dn receptor experiments, 'aggravated' compared to what?

Response:

We have clarified the text. Aggravated refers to an increased penetrance of adhesion defects beyond those expected from additive processes.

Line 226,226, should read Figure 2F

Response:

Corrected.

Line 227, "12-20% of tagged Ddr proteins were phosphorylated", need to refer to the data I cannot see that in Figure 2

Response: Has been clarified. Line 250 now reads "~12%" and refers to Figure 2g.

Line 228, "...suggesting that there is a large pool of inactive Ddr proteins within the cells." It is not clear how this conclusion can be made based on overexpression of tagged proteins. The authors should remove this statement.

Response:

Speculative statement was removed.

Line 234, should read Figure 2F, G

Response:
Corrected.

Line 254, should read Figure 3B, could not find a reference to Figure 3A

Response:
Corrected.

Line 265, that that

Response:
Corrected.

Line 445, dynamical (is not a word)

Response:
Corrected.

Response to Reviewer #2

The cardiac lineage of Ciona embryos represents the simplest example of collective cell migration with a pair of cardiopharyngeal progenitors (TVCs) undergoing a polarised migration following a well-characterised temporal sequence. In this study, the authors deploy an impressive panel of technically-challenging approaches to the Ciona system and analyse quantitative roles of Discoidin domain receptor (Ddr) and cell-matrix adhesion during the collective migration of TVCs.

The major claims of this study are as follows. 1) The TVC-specific Ddr, in cooperation with Integrin-b1, promotes cell-matrix adhesion to epidermis. At reduced levels of cell-matrix adhesion, the directionality of migrating TVCs is severely affected. 2) An endodermally-derived collagen, Col9-a1, is deposited in the matrix lining of the ventral epidermis and activates Ddr at the ventral membrane of migrating TVCs. 3) BMP-Smad signalling is differentially activated between the leading and tailing TVCs and Ddr promotes BMP-Smad signalling independently of its role in cell-matrix adhesion. Finally, 4) Ddr promotes Vegfr signalling in TVCs independently of its role in cell-matrix adhesion.

In contrast, cell-matrix adhesion and Vegfr signalling form an antagonistic relationship in TVCs. In order to highlight these findings succinctly, the authors integrate them in a simple schematic, which is presented as Figure 7.

As highlighted above, the current study involves major technical achievements. The successful application of Proximity Ligation Assay (PLA) to visualise a phosphorylated form of Ddr with subcellular resolution is impressive. I am also impressed with the cleverly-defined morphometric parameters in order to describe TVC migration quantitatively. The Christean lab also appears to have fully mastered the use of CRISPR/Cas9-mediated gene knockout in Ciona system

My major concern of this study is that some of the major claims are not statistically supported (i.e. with p-values). These claims are as follows:

Vegfr signalling negatively controls cell-matrix adhesion (Fig 5B)

Response:

While we do see a small increase in TVC surface in contact with the epidermis when we disrupt Vegfr signaling by overexpressing dnVegfr, it is difficult to quantitate because even cells that are fully flattened onto a surface would theoretically be able to devote only 50% of their surface to the contact. To circumvent this limitation, we show an inverse correlation between cell sphericity and contact with the epidermis (Figure 1h). We then include the data regarding the decreasing sphericity of dnVegfr expressing cells as supporting evidence (Figure 5c), suggesting that dnVegfr expressing TVCs are flatter than wild type TVCs we show.

**Cell-matrix adhesion negatively controls Vegfr signalling (Fig 5E,F),
Integrin-b1 promotes Ddr activation (Fig 2G),
Col9-a1 promotes Ddr activation (Fig 4C).**

Response:

We repeated our image analysis for PLA levels, which were done with an arbitrarily set threshold, using automated thresholding set by the Imaris data analysis software for PLA/HA signal detection and updated our analysis using two-way ANOVA to account for the leader/trailer cell differences. We find that the results are statistically significant.

Many of these claims are based on PLA experiments. It is also important that the authors describe in detail how they quantified the PLA results and expressed them as “% Phosphorylated Ddr/Vegfr/RTK”. I am wondering whether the results might be better represented by counting the number of PLA spots in the ventral and dorsal membranes and dividing it by total number of spots.

Response:

The full length receptors tagged with 3xHA which are required to perform the PLA are introduced into the embryo through electroporation, which results in variable levels of expression in any given embryo. Because of this we cannot reliably treat the number of PLA spots as total of the RTK because it is confounded by the total expression levels. We therefore normalize the PLA signal by the HA signal and report the proportion of HA that colocalizes with the PLA signal.

Similarly, there is no statistical analysis for Fig 1E,F and Fig 3D despite conclusions being drawn. I suggest that the authors include statistical support for all experiments where possible or clearly state when conclusions are hypothetical or why statistical analysis was not possible. Otherwise, it seems odd to see statistical analysis on some data but not on others.

Response:

We have re-evaluated our approach to this data and have now performed the appropriate statistical analyses, based on the structure of the data. Data in figure 1b and c are now analyzed using the chi-square test and standard error is plotted. New replicates have been added to figure 3d and two-way ANOVA was performed to determine statistical significance.

Minor points:

1) page 6/line 153, change Figure 1G to Figure 1E.

Response:

Corrected

2) page 8/line 225, change Figure 2E to Figure 2F.

Response:

Corrected

3) page 9/line 234, change Figure 2E,F to Figure 2F,G.

Response:

Corrected

Response to reviewer #3

This is an interesting report in which the authors investigated the mechanisms of collective polarity using migratory cardiopharyngeal progenitors of the tunicate *Ciona* as a model. Cardiopharyngeal progenitors (also termed as trunk ventral cells, TVCs) that polarize as leader and trailer cells, were found to express Discoidin domain receptor (Ddr) in a manner dependent on the transcription factor *Foxf*, a regulator of TVC migration. Ddr was found to co-operate with β 1-Integrin in TVCs, promoting cell adhesion to the epidermis, during TVC migration between the trunk endoderm and ventral epidermis. Endoderm-derived collagen 9-a1 (*Col9-a1*) was deposited into the basal epidermis and was found necessary for activation of Ddr at the TVC-epidermis interface. *Vegfr* was found to antagonize Ddr/ β 1-Integrin-mediated TVC adhesion to the ventral epidermis. Furthermore, Ddr regulated leader-trailer-polarized BMP-Smad signaling independently of its role in cell-matrix adhesion. Collectively, the authors propose a model where Ddr, acting downstream of cardiopharyngeal-specific transcription factors, regulates both collective polarity and directed TVC migration/adhesion.

1. The authors should report which statistical tests were used, as well as the number of independent experiments/embryos (n) in each figure. Since much of the data is based on quantitative image analysis, details on these should be added (e.g. is each data point derived from a different embryo).

Response:

Description of statistical analysis is now included in each figure. All relevant statistically significant differences are given. We have clarified the detail in each figure legend.

2. The authors suggest that BMP signalling is enhanced in the *Col9-a1* mutants (loss-of-function by CRISPR/Cas9), due to increased bioavailability of BMPs. Does *Col9-a1* (type IX collagen) bind BMPs? *Col9-a1* is very different from collagen IV that is known to bind BMP2/4, thus, it may not be plausible to assume that they function in a similar manner.

Response:

We are grateful to the reviewer for calling our attention on this matter. We have moved this speculation to the discussion with the appropriate references.

3. The authors conclude that Col9-a1 is the ligand for Ddr, since Col9-a1 deletion decreased Ddr phosphorylation (Fig.4). However, have the authors examined if loss of Col9-a1 affects deposition, organization or stability of other matrix components, thereby excluding potential indirect effects due to Col9-a1 loss that might prevent Ddr activation in their model? As suggested for this class of non-fibrillar collagens, type IX collagen is expected to link type II collagen fibers and the proteoglycans, thus contributing to the overall matrix organization in the cartilage.

Response:

We thank the reviewer for this insightful comment. We have modified the text and summary model to keep open the possibility that Col9-a1 impacts Ddr activity indirectly, via an effect on the collagen ECMs. We are interested in learning more about the structure and function of the extracellular matrix in Ciona, although Ciona embryos do not contain cartilage, but currently lack the molecular tools to address this fully and their generation would take a considerable amount of time. We therefore reserve this for the discussion and will address this in future studies.

Which collagens activate Ddr in Ciona, are these still expressed in the mutant embryos and does the ligand specificity differ between Ciona and human Ddr?

Response:

Currently, we do not know which collagens bind Ddr directly, nor the effect that the loss of Col9-a1 has on other collagen expression. Ciona embryos express multiple collagens (Sasakura et al., Dev Genes Evol (2003) 213:303–313; and our unpublished observations) and it is possible that Ddr responds to these collagens as well. It seemed to us that this potentially fascinating work is beyond the scope of this paper.

Col9-a1 staining is necessary to show the decrease in Col9-a1 expression in mutants vs wild type embryos.

Response:

Currently, there is no Col9-a1 antibody available that has reactivity to the Ciona Col9-a1. We have now added peak-shift analysis of the CRISPR-targeted Col9-a1 locus in supplemental figure 5 as confirmation of CRISPR efficiency.

4. Fig. 2F, 3C and 4C show an almost complete loss of phospho-Ddr, whereas the quantifications in 2G, 3D and 4C are less striking, and lack significance – why? The authors should consider referring those results as a “trend”, and modify the figure titles in Fig 3 and Fig 4 accordingly

Response:

We have replaced the micrographs in the figures mentioned above with images more representative of our quantitation results. Additional statistical analysis using two-way ANOVA has been performed and any statistical significance is indicated in the figures.

5. The asymmetric exposure of TVCs to the endodermal derived Col9-a1 was concluded to stimulate leader/trailer polarization via BMP-Smad signalling (Fig. 6). Was

polarization of TVCs affected/rescued in Nkx2-1>dnSar1 embryos with epidermal expression of EfnB>Col9-a1::GFP?

Response:

This is a great question, and very tantalizing hypothesis although preliminary observations did not indicate an effect of Col9-a1::GFP misexpression on collective polarity and we did not pursue this further.

6. In fig. 5, dnVegfr was found to partially suppress the de-adhesion phenotypes produced by dnDdr or dnInt β 1 in the TVCs, and Col9-a1 inhibition and dnInt β 1 misexpression resulted in increased of Vegfr activity (Figure 5), suggesting that cell adhesion via β 1-Integrin negatively regulates Vegfr activity. However, since dnDdr decreased Vegfr activity, other factors must be at play.

Response:

This is very well observed. Indeed these effects are incongruous at first sight, and we agree with the reviewer that other effects must be invoked. For instance, such opposite and adhesion-independent effects of Col9-a1 and Ddr on Vegfr might be mediated by BMP-Smad signaling, which responds to these perturbations in qualitatively similar ways. While testing such hypotheses is feasible, we reasoned that these new experiments would further stretch the paper and extend beyond reasonable limits.

Potential candidates affecting Vegfr activity would include ligand or phosphatase-mediated regulation of Vegfr; have the authors considered any of these and can these be discussed?

Response:

We have analyzed Vegf expression in the Ciona embryo by in situ hybridization but have not detected transcripts in embryos, suggesting that Vegfr may be activated via a non-canonical, perhaps ligand-independent, pathway(s).

7. Determining the α -integrin subunit that pairs with β 1-integrin would strengthen the authors' conclusions about the importance of Col9-a1.

Response:

We have assayed the expression of multiple alpha-integrin subunits but have not been able to identify a TVC-specific candidate.

Minor comments:

8. Fig 3A and Fig S5 were not referred to in the text.

Response:

Figures have been updated and are referenced in the text.

9. Results in Fig S3, demonstrating the effects of Ddr, Vegfr and β 1-Integrin on leader-trailer cell surface contact area are not discussed in the text.

Response:

This data has been removed from the manuscript in order to focus it on the main points, which are already numerous.

10. Details of the EfnB>Col9-a1::GFP construct are missing in the methods.

Response:

Additional information has been added to the materials and methods. Primers used for amplification and subcloning of the Col9-a1 coding domain are given in supplemental table 1.

11. Color coding in Fig S6 is missing.

Response:

Color-coding has been added. Figure S6 is now figure S7.

Reviewers' comments:

Reviewer #1 (Remarks to the Author):

In this revised manuscript the authors have conducted additional experiments and improved their statistical analysis. The paper is interesting and examines the role of the Discoidin domain receptor (RTK) Ddr in cardiopharyngeal progenitor cells in detail. The findings are original, focussing on *Ciona*. Some discussion regarding the relevance to migrating cardiac progenitor cells in vertebrates might be of interest.

A remaining issue is that the number of embryos used for the live imaging experiments is limited, which is problematic given that the observed effects are very small.

The authors continue to make some strong statements in the results section, these should be amended/removed when they are poorly supported. For example, the very generalized statement regarding TVCs expressing DdrDN showing tumbling behaviour is not justified. This is apparently only being observed in 15% of embryos imaged, given that an n=7 embryos were imaged, this would equate to only 1 embryo. How can they be confident that this is biologically relevant?

Line 149: "TVCs expressing Foxf-driven Ddrdn retained their ability to initiate migration but the cells fail to maintain relative leader/trailer positions and followed more variable migration paths, resulting in a tumbling motion in 15% (n=7) of embryos imaged (Figure 1d,e Movie S2).

Similarly, the statement on (line 162) "Vegfrdn expression increased track straightness" (Figure 1g). I cannot see that, the track straightness is indicated by the slope of the line and that seems to be almost identical (blue vs black).

The sphericity difference between leader and trailer appears to be 0.81 vs 0.85, the surface contacts are approximately 45-50% (Leader) or 40-45%, (trailer) – again, is that biologically relevant? It is not clear how the detachment phenotypes shown in Fig. 2d were classified as either mild, moderate or severe. The grading is not explained nor if this was done blind by the assessor(s).

Line 282 should read Figure 3c

The authors now show the design of two guide RNAs targeting Col9-a1, but still do not demonstrate that gene editing was successful or that Col9-a1 expression/function was indeed lost in TVCs.

Reviewer #2 (Remarks to the Author):

In this revised manuscript, the authors have addressed satisfactorily all the points I raised. As I expressed in the first round of reviewing, the authors have deployed an impressive panel of technically-challenging approaches to the *Ciona* system. The manuscript is methodologically sound and its major claims are now statistically supported.

Reviewer #3 (Remarks to the Author):

The authors have answered most of my comments; however, few minor issues remain to be considered.

The authors have improved on the statistical analysis, and included the statistical comparisons for

most figures, except for Fig. S4b-c. The authors should report the statistical significance also for these results. If not significant, the results should be referred to as a trend (p. 8, starting from row 196).

For statistical analysis of phospho-proteins in Figs. 3e, 4c, 5d and 5f, please indicate in the legend if data was pooled from both L and T cells to compare phospho-protein levels between treatments.

Fig. 1g. The significance of this analysis remains unclear in the absence of information about the goodness of fit of the linear model used to model the data.

Row 99. Replace "vascular endothelium growth factor receptor" with "vascular endothelial growth factor receptor".

Fig. S2b. Please include the time points in the Fig. S2b.

Reviewer #1 (Remarks to the Author):

In this revised manuscript the authors have conducted additional experiments and improved their statistical analysis. The paper is interesting and examines the role of the Discoidin domain receptor (RTK) Ddr in cardiopharyngeal progenitor cells in detail. The findings are original, focussing on Ciona. Some discussion regarding the relevance to migrating cardiac progenitor cells in vertebrates might be of interest.

A remaining issue is that the number of embryos used for the live imaging experiments is limited, which is problematic given that the observed effects are very small.

The authors continue to make some strong statements in the results section, these should be amended/removed when they are poorly supported. For example, the very generalized statement regarding TVCs expressing DdrDN showing tumbling behaviour is not justified. This is apparently only being observed in 15% of embryos imaged, given that an n=7 embryos were imaged, this would equate to only 1 embryo. How can they be confident that this is biologically relevant?

Line 149: “TVCs expressing Foxf-driven Ddrdn retained their ability to initiate migration but the cells fail to maintain relative leader/trailer positions and followed more variable migration paths, resulting in a tumbling motion in 15% (n=7) of embryos imaged (Figure 1d,e Movie S2).

Response:

We have rephrased the text to emphasize that these were observations, and stop short of claiming a general phenomenon.

However, it is important to consider that these detachment phenotypes, while observed on a small number of live embryos, are consistent with our observations on >50 fixed embryos for associated changes in cell shape and surface contact. This is illustrated in figure 2a,c, where the total percentage of embryos where the TVCs no longer contact the epidermis with either leader or trailer is 15%. In this case our reported percentage of total detachment is in agreement with our reported percentage for the tumbling phenotype observed during live imaging. Thus, the observed tumbling behavior is probably biologically relevant as a consequence of inhibited cell-matrix migration, which is our main conclusion. In other words, we use the live imaging experiments as a starting point to further investigate regulation of TVC interaction with surrounding tissues, and we report observations, but the conclusions are grounded on a more extensive bundle of evidence.

Similarly, the statement on (line 162) “Vegfrdn expression increased track straightness” (Figure 1g). I cannot see that, the track straightness is indicated by the slope of the line and that seems to be almost identical (blue vs black).

Response:

Here too, we are reporting the observations, and do not attempt to directly interpret them and conclude about the biological process from single observations. Our proposed model for the interaction between Ddr and Vegfr results from a more extensive bundle of evidence. Specifically, we observed an 11% increase in track straightness for Vegfr^{dn} compared to wild type. As this difference was not statistically significant, we have changed the wording in line 163 to reflect that this is a trend highlighting a potential role for Vegfr in TVC migration.

The sphericity difference between leader and trailer appears to be 0.81 vs 0.85, the surface contacts are approximately 45-50% (Leader) or 40-45%, (trailer) – again, is that biologically relevant?

Response: Here too, these are our observations, and we report them in a rather matter-of-fact manner. The exact interpretation and conclusions with regards to the collective polarity goes beyond these exact numbers, and we do not attempt to stretch these interpretations. For instance, we present additional evidence for leader-trailer polarity, and these are some of the parameters. These differences may be subtle, but could still be “biologically relevant”.

For instance, the differences were statistically significant, and probably reflect the morphological dynamics of collectively migrating cells, where the leader establishes new adhesions upon producing and stabilizing lamellipodia, whereas the trailer markedly detaches from the rear end.

It is not clear how the detachment phenotypes shown in Fig. 2d were classified as either mild, moderate or severe. The grading is not explained nor if this was done blind by the assessor(s).

Response:

We explained the criteria for classification in Supplemental Figure 3. Here too, these analyses are meant to complement other observations reported in the paper, and we want to stress that our main conclusions do not stem from any single individual observations, but rather from a consistent set of observations, and we stand by the main conclusions of the paper.

Line 282 should read Figure 3c

Response:

Line 282 has been corrected.

The authors now show the design of two guide RNAs targeting Col9-a1, but still do not demonstrate that gene editing was successful or that Col9-a1 expression/function was indeed lost in TVCs.

Response:

Col9-a1 expression is depleted from the overlying endoderm, where it is normally expressed - not the TVCs- using the early anterior vegetal hemisphere driver *Foxd* to express Cas9, which target the endoderm progenitors. We have now added additional validation of the loss of Col9-a1 expression under Col9-a1 CRISPR conditions by using in situ hybridization to detect loss or reduction of Col9-a1 expression in the endoderm. We find that approximately 45% of embryos lose or reduce Col9-a1 expression under Col9-a1 CRISPR conditions, this is shown in Figure S5c. Targeting of the Col9-a1 locus in the endoderm also produces a specific TVC detachment phenotype which phenocopies the defects seen when disrupting TVC adhesion or secretion from the endoderm. We therefore conclude that the observed phenotypes can be attributed to loss of Col9-a1 function in the endoderm.

In general, we have extensively characterized the effects of CRISPR/Cas9-mediated loss-of-gene function in *Ciona* embryos, developed the methods used to achieve efficient gene knock-down/out (Stolfi et al., Development, 2014; Gandhi et al., Dev Biol, 2017), and people in the field have turned to us, or directly used our constructs as made available on Addgene, so we are pretty confident about our ability to perform lineage-specific gene loss-of-function using CRISPR/Cas9 in *Ciona* embryos.

Reviewer #2 (Remarks to the Author):

In this revised manuscript, the authors have addressed satisfactorily all the points I raised. As I expressed in the first round of reviewing, the authors have deployed an impressive panel of technically-challenging approaches to the *Ciona* system. The manuscript is methodologically sound and its major claims are now statistically supported.

Reviewer #3 (Remarks to the Author):

The authors have answered most of my comments; however, few minor issues remain to be considered.

The authors have improved on the statistical analysis, and included the statistical comparisons for most figures, except for Fig. S4b-c. The authors should report the statistical significance also for these results. If not significant, the results should be referred to as a trend (p. 8, starting from row 196).

Response:

Wording in line 198 has been changed to reflect that the effect is an observed trend.

For statistical analysis of phospho-proteins in Figs. 3e, 4c, 5d and 5f, please indicate in the legend if data was pooled from both L and T cells to compare phospho-protein levels between treatments.

Response:

The data used in these graphs was analyzed separately for leaders and trailers, the cell type is indicated below the graphs.

Fig. 1g. The significance of this analysis remains unclear in the absence of information about the goodness of fit of the linear model used to model the data.

Response:

Compared to the initial version, we chose to represent the distance migrated and the track length to show the measurements underlying the calculation of track straightness, which is -by definition- the slope of the curves represented. In other words, we could calculate track straightness from each embryo and then average the values (as shown in Supplemental Figure 2d). In this regard, the linear regression is not the essence of this analysis and, indeed, r^2 values (~goodness of fit) are low because the measured values are variable, especially in experimental perturbations, which is consistent with the variability of migration paths described in Fig. 1e. Moreover, the line is forced to intercept the origin (again, to recapitulate the average straightness), which cause the r^2 values to be negative. This data still illustrates a trend connecting the functions of Ddr and Vegfr to the directionality of TVC migration. We added the r^2 values to provide a more extensive description of these variable phenotypes (i.e. individual measurements of straightness) in supplemental figure 2d.

As noted in response to other comments above, this panel is very much a matter-of-fact report of observations and the main conclusions, and thus the “significance of these analyses”, emerged from further analyses presented in subsequent figures (i.e. Figures 5 and 6 in the case of Vegfr and collective polarity).

Row 99. Replace “vascular endothelium growth factor receptor” with “vascular endothelial growth factor receptor”.

Response:

Line 99: Corrected.

Fig. S2b. Please include the time points in the Fig. S2b.

Response:

Time points have been added to Supplemental Figure 2b.

REVIEWERS' COMMENTS:

Reviewer #1 (Remarks to the Author):

The authors have addressed my remaining queries satisfactorily and either made minor changes or provided clarification.

Reviewer #3 (Remarks to the Author):

The authors have made satisfactory adjustments to the manuscript in response to most of the comments.

However, since r^2 values in Fig. 1g are negative, the information of this graph remains low. Thus, the data might be moved into the supplement. As stated by the authors in their response, the figure illustrates a trend, and this should be clearly indicated in the text on p. 6 and 7. The authors explain (in the legend for Fig. 1g) that "the line is forced to go through the origin, reflecting that cells that fail to move have a total displacement = 0 and track length = 0.". However, it should be possible (at least in theory) that track length is >0 , even though total displacement = 0.

Reviewer #1 (Remarks to the Author):

The authors have addressed my remaining queries satisfactorily and either made minor changes or provided clarification.

Response

We thank the reviewer for their helpful comments.

Reviewer #3 (Remarks to the Author):

The authors have made satisfactory adjustments to the manuscript in response to most of the comments.

Response

We thank the reviewer for their helpful comments.

However, since r^2 values in Fig. 1g are negative, the information of this graph remains low. Thus, the data might be moved into the supplement. As stated by the authors in their response, the figure illustrates a trend, and this should be clearly indicated in the text on p. 6 and 7. The authors explain (in the legend for Fig. 1g) that “the line is forced to go through the origin, reflecting that cells that fail to move have a total displacement = 0 and track length = 0.” However, it should be possible (at least in theory) that track length is >0 , even though total displacement = 0.

Response

The reviewer is absolutely correct that a cell could have a total displacement of 0 while still retaining its motility. However, our data for the graph in figure 1g comes from live imaging of TVC migration, which would allow us to differentiate between cells that fail to move and cells that move in a circle. We have not detected such events in any of our live imaging sets, suggesting that under the conditions we tested all TVCs retain their competence to migrate. We feel that this graph provides a good overview of observed cell behavior and therefore elect to keep it in figure 1.